

# Rapid attribution of the May/June 2016 flood-inducing precipitation in France and Germany to climate change

Geert Jan van Oldenborgh[1], Sjoukje Philip[1], Emma Aalbers[1], Robert Vautard[2], Friederike Otto[3], Karsten Haustein[3], Florence Habets[4], Roop Singh[5], and Heidi Cullen[6]

[1]Royal Netherlands Meteorological Institute (KNMI), De Bilt, Netherlands
[2]LSCE/IPSL Laboratoire CEA/CNRS/UVSQ, Gif-sur-Yvette, France
[3]Environmental Change Institute, University of Oxford, Oxford, U.K.
[4]CNR/Pierre & Marie Curie University, Paris, France
[5]Red Cross Red Crescent Climate Centre, The Hague, Netherlands
[6]Climate Central, Princeton, U.S.

*Correspondence to:* G. J. van Oldenborgh (oldenborgh@knmi.nl)

**Abstract.** The extreme precipitation that would result in historic flooding across areas of northeastern France and southern Germany began on May 26th when a large cut-off low spurred the development of several slow moving low pressure disturbances. The precipitation took different forms in each country. Warm and humid air from the south fueled sustained, large-scale, heavy rainfall over France resulting in significant river flooding on the Seine and Loire (and their tributaries), whereas the rain came

from smaller clusters of intense thunderstorms in Germany triggering flash floods in mountainous areas. The floods left tens of thousands without power, caused over a billion Euros in damage in France alone, and are reported to have killed at least 18 people in Germany, France, Romania, and Belgium. The extreme nature of this event left many asking whether anthropogenic climate change may have played a role. To answer this question objectively, a rapid attribution analysis was performed in near-real time, using the best available observational data and climate models.

In this rapid attribution study, where results were completed and released to the public in one week and an additional week to finalise this article, we present a first estimate of how anthropogenic climate change affected the likelihood of meteorological variables corresponding to the event, 3-day precipitation averaged over the Seine and Loire basins and the spatial maximum of 1-day precipitation over southern Germany (excluding the Alps). We find that the precipitation in the Seine basin was very rare in April–June, with a return time of hundreds of years in this season. It was less rare on the Loire, roughly 1 in 50 years in

April–June. At a given location the return times for 1-day precipitation as heavy as the highest observed in southern Germany is 1 in 3000 years in April–June. This translates to once roughly every 20 years somewhere in this region and season.

    The probability of 3-day extreme rainfall in this season has increased by about a factor 2.3 (>1.6) on the Seine a factor 2.0 (>1.4) on the Loire, with all four climate models that simulated the statistical properties of the extremes agreeing. The observed trend of heavy 1-day precipitation in southern Germany is significantly negative, whereas the one model that has the

correct distribution simulates a significant positive trend, making an attribution statement for these thunderstorms impossible at this time.



## 1   Introduction

From 26 May to 4 June 2016, a low pressure system was almost stationary over France and Germany. Due to their differing locations relative to the weather system's center, France and Germany underwent heavy rainfalls with different characteristics: moderate but continuous rain, partially large-scale, partially convective, over 3 consecutive days were reported in Central and North-East France while severe thunderstorms hit Southern Germany.

In France, this precipitation came on top of an already wet spring season. As a consequence, flooding first on smaller rivers like the Yvette and Loing, and later high water levels in the Loire and especially the Seine were reported. The highest 3-day rainfall occurred on May 29–31, see Fig. 1a.

The maximum amount of rain fell on the center of the Loire basin around the cities of Orléans and Tours, leading to the flooding of highways on May 31th with numerous drivers being stranded. The famous Chambord castle was flooded on June 1st. The floods mostly affected small tributaries with reduced warning systems.

The Loing river, a tributary of the Seine basin, reacted promptly to the heavy rainfall and several cities were flooded on June 1st, with more than 4000 people evacuated at Nemours. It is the most severe event ever reported on the Loing basin. Although only representing less than 9% of the Seine basin, the Loing river contributed to around 25% of the flood peak of the Seine. The Yvette river, a small tributary in a heavily urbanised area southwest of Paris flooded several cities on the morning of June 2nd, resulting in more than 2000 evacuees in Longjumeau. The Seine crested in Paris on June 3rd at a height above 6.1 m (some measurements problems occurred during the peak). This was not a record on the Seine river, the water level was lower than during the major flood of 1910 (and also lower than the floods in 1924, 1945, 1955, 1959 and 1982). It is estimated to have a return period of about 20 yr. There was considerable damage upstream of Paris, with four fatalities, thousands of evacuees, and an economic cost estimated above €1 billion (Telegraph, 5 June 2016). Estimates indicate that 828,100 inhabitants are exposed to flood in the Île-de-France, mainly because buildings precede regulations that prohibit building in floodplains (Faytre, 2011).

However, the damages were less in Paris itself. A timely training of crisis managers conducted in Paris in March 2015, Operation Sequana, simulated a major flood in the Île-de-France over eleven days and ensured authorities were prepared (www.prefecturedepolice.interieur.gouv.fr/Sequana/EU-Sequana-2016). Following their mandated protection plan, the famed Louvre museum evacuated 250,000 precious artworks from underground store rooms to upper floors of the museum (Washington Post, 2 June 2016), a precaution that proved unnecessary. Essential networks that are typically vulnerable to flooding due to their underground location (phone, railway, drinking and sewage) were largely unaffected in Paris, although the Metro line that ran directly along the Seine was forced to close and many people were left without electricity.

A multi-reservoir system exists to reduce floods and sustain low flows in the Île-de-France. The four reservoirs in the system are operated independently and follow filling curves that determine the target amount of water retained in the reservoir each day of the year. During the high flow seasons of November - June they store water in order to maintain low flows during the upcoming dry season, a necessary function to regulate water levels for shipping and drinking water to approximately 20 million people, among other socio-economic benefits (Ficchì et al., 2016; Dorchies et al., 2014). During this flooding event, all of the reservoirs were near 90% full in anticipation of typically dry conditions in the summer months. Large floods in late spring are





rare as virtually all Seine floods occur in winter. In the historical record only July 1659 and June 1856 recorded floods outside of the extended winter (December–March), and the observed river flow on June 2nd 2016 in Paris was 46% above the previous records in June available since 1886. The defences currently in place provide effective protection against frequent small floods and significantly lower the risks of larger floods and the associated damage. However, they are vulnerable to spring flooding when reservoirs are typically almost full in anticipation of drier conditions, as well as to successive flooding events that the dams may lack capacity to moderate (Roche, 2004). In addition, while all four reservoirs are located in the upstream areas of the Seine basin, some of the smaller tributaries that encountered the worst flooding during this event, are located outside of the reservoir catchment areas. Despite this, the reservoirs are estimated to have reduced the flood peak in Paris by about 0.23m ($-40\,\mathrm{m}^3\mathrm{s}^{-1}$, Seine Grands Lacs, 2016).

The meteorological variable that corresponds to the floods in France was taken to be the 3-day precipitation averaged over the river basins. Based on the basin seizes, we estimated that this time scale is close to the response time of the rivers. It was also the most exceptional. We analysed both the Seine and the Loire. Maps of these basins are shown in Fig. 2a.

This study is an attribution analysis of the heavy rainfall event that occurred over 29–31 May, and not a full attribution analysis of the associated floods themselves. Therefore although heavy rainfall as a variable captures a relevant aspect of the flooding events which occurred, many other contributing factors are neglected in this rapid attribution study. Firstly the soil types and saturation levels at the time of this extreme rainfall event have not been captured. Due to a wet spring and the rainfall in the few days preceeding 29 May, saturation levels limited the absorption capacity of additional rainfall. This analysis also does not take into account the impacts of the reservoirs including their location upstream, and their holding capacity at the time of the event. In addition, land cover and associated runoff characteristics have also not been taken into account. A full attribution of the floods themselves, rather than just the rainfall event, would need to take all of these factors into account.

In Germany, the effects of the low-pressure system were very different. There it induced a series of very heavy thunderstorms, sometimes organised in mesoscale systems. These caused flash floods in many locations, e.g., the widely shown devastation in Braunsbach in Baden-Württemberg (South Germany) in the night of 29–30 May. On 1 June there was widespread flooding in the southeastern corner of Germany (Niederbayern), with seven fatalities.

The highest reported 1-day rainfall amount in the period preceding the floods is 122 mm/day in Gundelsheim, Baden-Württemberg, which fell between 29 May 6.00am UTC and 30 May 6.00am UTC according to the German Weather Service, DWD. The highest 1-day precipitation in May 2016 is plotted in Fig. 1b, based on the 0.25° E-OBS analysis, showing multiple maxima over Germany. It should be noted that the precipitation typically fell in a few hours in these events, much shorter than one day, which was confirmed by radar observations of the German Weather Service that show more than 90 mm/hr in the area of Braunsbach.

However, the data required to analyse the event at the sub-daily scale in real time is not yet available to us (although it is publicly available at DWD). As a measure corresponding to the most severe impacts we took the highest local 1-day precipitation in central and southern Germany, 48–51°N, 7–13°E in the models. The observed maximum precipitation in the station data in the period 1951-now is shown in Fig. 2b. The area that is covered by the observation data extends slightly further to the east to capture a few stations with heavy flooding.





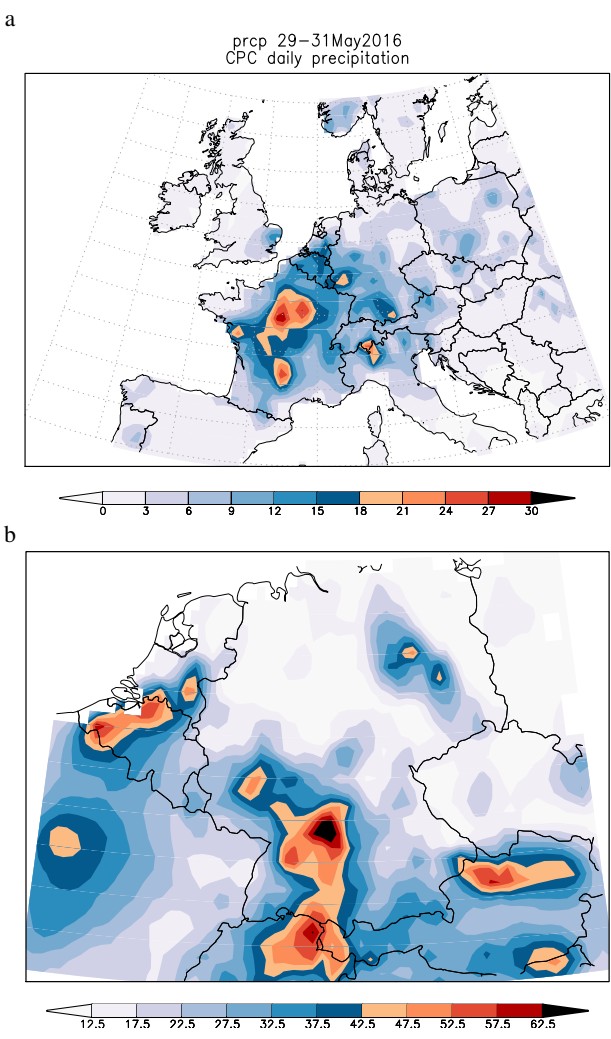

**Figure 1.** a Precipitation averaged over 29–31 May 2016 (mm/dy). b Highest 1-day precipitation in May 2016 (mm/dy). Source: a NOAA/NCEP/CPC, b E-OBS.





a

b

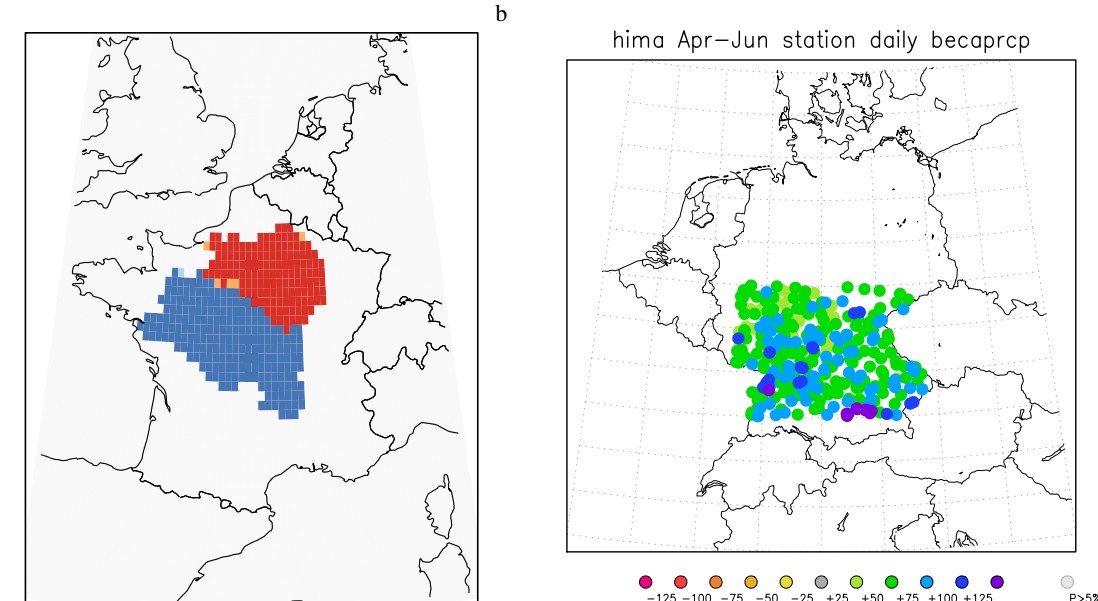

**Figure 2.** a Maps of the Seine (red) and Loire (blue) basins on the E-OBS grid, b Maximum observed precipitation in the period 1951-now, in the region 48–51°N, 7–13.5°E, recorded at weather stations in the ECA&D dataset (mainly from DWD). A few stations further east than the model analyses, which end at 13°E, were included.

The attribution of trends in heavy precipitation to human influence has also been a focus in recent years (e.g., Pall et al., 2011; Schaller et al., 2016). These pioneer studies showed the potential of providing statements about the role of human activities on weather extremes. However demand on such information is often in real time, when, for a couple of days, damages and losses raise the attention of the public and media. A challenge is therefore to provide scientifically sound and reliable information 5 in near real time (about a week) about human influence on extreme events. This was first accomplished for the winter storm Desmond by van Oldenborgh et al. (2015).

Considering events similar to the rain studied here, Schaller et al. (2014) found no trend in 3-day rainfall averaged over the upper Elbe and Danube basins in the same late spring/early summer season, whereas Vautard et al. (2015) showed a significant increase of about 5%/decade in extreme convective autumn rainfalls in a Mediterranean mountain range in southern France 10 over the past 60 years.

On a coarser scale, van den Besselaar et al. (2013) studied trends in 1 in 20-year events over 1951-2010, where they divide Europe in a northern and a southern half at 48°N. They found that these events become more common in northern Europe in spring, but show no significant change in southern Europe, both for 1-day and 5-day periods.

In projections, Nikulin et al. (2011) investigated projected changes in 20-year return time precipitation events in a set of 15 RCA3 regional climate model simulations driven by six different global climate models. The projected future changes in summertime extreme precipitation for the individual simulations show mixed small-scale changes with, on average, a tendency to an increase in northern and a decrease in southern Europe. Rajczak et al. (2013) used a set of regional climate models in order





to study projections of climate change on heavy precipitation events in Europe from 1970–1999 to 2070–2099. They study, among others, return values of 5-day and 1-day precipitation intensity with a return period of 5 years. For France they find that the 5-day precipitation is increasing in spring and decreasing in summer. For Germany the 1-day precipitation is increasing in both spring and summer. However, the region with decreasing trends in southern Europe is close to Southern Germany. Note

that the events we are investigating in this study have much higher return times.

Jacob et al. (2014) also studied the evolution of several indicators, using the recent EURO-CORDEX regional climate projection ensemble and scenarios and found significant increases in heavy rainfall for the period 2071–2100 under the RCP8.5 scenario.

In this article, we report the results of a rapid attribution of the French and German 2016 events, a study that was carried out

through collaboration of several organization in a time period of less than 10 days, using well-established techniques. The trends in the above quantities, 3-day basin averaged precipitation in France, 1-day area-maximum in Germany, are investigated using comparisons with either the climate around 1960 or in a counterfactual climate without anthropogenic emissions, described in the methods section. The analysis of observations and five model ensembles are described in the next sections. This is followed by a synthesis in the discussion section and conclusions.

## 2   Methods

First we evaluate whether the observational analyses and models represent the statistics of high spring precipitation well enough to be able to use them. We do this mainly by fitting the 3-daily or daily extremes in April to June to a Generalised Extreme Value (GEV) distribution (Coles, 2001), which is assumed appropriate for these block maxima. To account for possible changes we scale the distribution with a measure of climate change, for which we take the 4-yr smoothed global mean temperature.

Specifically, we take

$$
\begin{aligned}
F(x) &= \exp\left[-\left(1+\xi\frac{x-\mu}{\sigma}\right)^{1/\xi}\right], \\
\mu &= \mu_0\exp(\alpha T'/\mu_0), \\
\sigma &= \sigma_0\exp(\alpha T'/\mu_0),
\end{aligned}
\tag{1}
$$

such that the ratio $\sigma/\mu$ is constant. The fit is performed using a maximum likelihood method varying $\alpha, \mu_0, \sigma_0$ and $\xi$, with an

added penalty term on $\xi$ with a width of 0.2 so that values larger than about 0.4 are penalised as unphysical. Uncertainties are estimated with a 1000-member non-parametric bootstrap. We take all years to be independent but take correlations between neighbouring stations or similar ensemble members into account with a moving block technique.

The three parameters of the distribution are the location parameter $\mu$, the scale parameter $\sigma$ and the shape parameter $\xi$. We evaluate these for the year 2016 and compare the values for the observational estimates (analyses and reanalysis). Models

are evaluated demanding that these parameters are similar to the distribution fitted to the longest observational analysis, if necessary after a multiplicative bias correction.





The next step is trend detection. For the observed record and reliable models for which simulations of the historical record are available, we fit Eq. 1 to the maxima. Inserting the years 1960 (or the first year with data if later) and 2016 in Eq. 1 gives the probability for the event in these years, $p_0$ and $p_1$. These are expressed as return times $\tau_i = 1/p_i$. The ration of these is commonly referred to as the risk ratio, $\mathrm{RR} = p_1/p_0 = \tau_0/\tau_1$. This indicates how much more likely the event is now than in 1960, but does not attribute this difference.

To attribute the change we use two models that also have experiments simulating a counterfactual world without anthropogenic emissions of greenhouse gases and aerosols. These allow us to compute how much more likely or unlikely the event has become due to these emissions. Often we can neglect the effect of natural forcings on these extremes. In that case the change from 1960 to now is about two-thirds of the change from pre-industrial conditions to now (about $1°$ C warming).

# 3   Observational analysis

We compared the 0.25° E-OBS analysis 1950-now (Haylock et al., 2008) and 0.5° CPC analysis 1979-now http://www.cpc. ncep.noaa.gov/products/Global_Monsoons/gl_obs.shtml and for Germany the DWD station data obtained via ECA&D. As French precipitation data were not available in real time, the analyses there are based on a relatively sparse subset of stations. We checked that the decorrelation scales of 3-day precipitation are large enough that this is not a problem by comparing with the ERA-interim reanalysis for the past. Satellite-derived products do not perform well in this region and season.

The CPC analysis, which is updated daily, shows the highest sums over 29–31 May 2016: the 3-day sum was 55,mm/3dy averaged over the Seine basin and 47 mm/3dy for the Loire basin, see Fig. 3. These values were taken to represent the observed event in the following as the E-OBS data for 29–31 May 2016 was not yet available.

For the 3-day and basin-averaged precipitation the distributions of the CPC and E-OBS analyses are remarkably similar, even though they cover different time periods (see Tables 2–3). The ERA-Interim reanalysis also resembles these well, albeit with a slightly smaller scale parameter. The trend analysis using Eq. 1 shows that the return time is larger than can be determined with the longest series, E-OBS: $\tau_1 > 150$ yr for such an event in April–June. The time series from 1950, Fig. 4a, shows how unusual this amount is for late spring/early summer. The best fit for the trend is positive. However, the uncertainties are large and easily encompass zero.

The model analyses in the next sections show that for higher statistics the shape parameter $\xi$ is very close to zero. We can improve the estimate of the return time by setting $\xi=0$, i.e., assuming a Gumbel distribution. This gives a best estimate of 180 yr, with still a wide 95% uncertainty range of 50 to 3000 yr.

The 3-day rainfall in the Loire basin was less exceptional (Figs 4bd), with a return time of the order of 100 yr in April–June (2.5% lower bound 15 yr). The observations also show a positive trend that is not significantly different from zero. A Gumbel fit gives a return time of 35 yr (10. . . 300 yr).

For Germany we first use station data from the DWD, obtained via the ECA&D dataset. We selected all stations in the study area (extended to 13.5°E) with at least 50 years of data, giving 247 stations. The number of active stations increases sharply to 230–240 from 1951 to 2000 and decreases to about 190 recently. Due to this decrease taking the maximum value across all





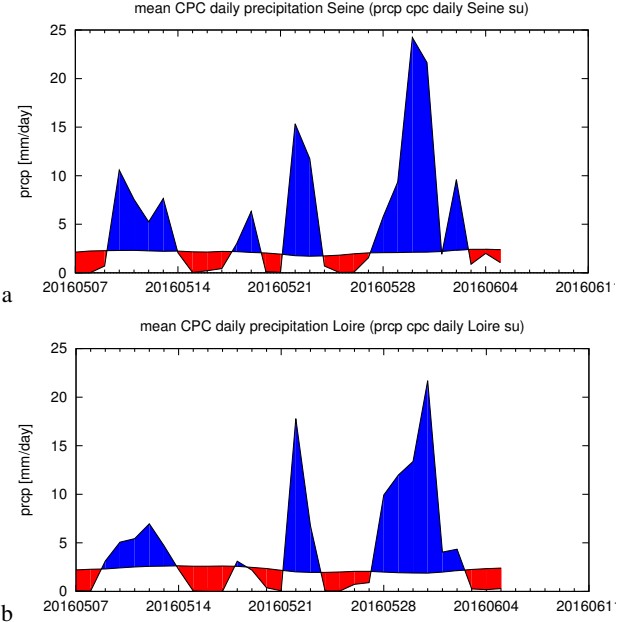

**Figure 3.** CPC analysed sums of precipitations averaged over the Seine (a) and Loire (b) basins.

stations would introduce a spurious downward trend, as the probability of an extreme shower hitting a rain gauge decreases over time. We therefore analyse all April–June maxima together as in Vautard et al. (2015), taking spatial correlations into account with a moving bootstrap procedure and starting in 1951 to minimise the inhomogeneity due to the start of the modern network. The effect of the assumption that all stations are identically distributed is checked by redoing the calculation with all
5 series normalised to a common mean.

The results are shown in Fig. 5. The return time of 122 mm/dy in the present climate is about 3000 years in Apr–June for a single station. The probability of observing this at any station in the region is of course much higher, 23 yr (8 to 1800 yr), as these showers are small compared to this area. The set of stations shows a negative trend: the probability of observing such an extreme daily sum is now 0.6 times what it was in 1960 (0.4 to 0.7). Normalising all series to the same mean gives a RR of 0.5
(0.4 to 0.8), so the spatial heterogeneity does not affect this result.

We also fitted the spatial maximum of E-OBS to Eq, 1. This dataset has 30% lower extremes due to the gridding with a decorrelation scale that is smaller than the size of a thunderstorm. The ration $\sigma/\mu$ and the shape parameter $\xi$ agree with the station data. This fit gives a return time in 2016 of 9 to 300 yr, in agreement with the station analysis. The trend is positive but not significantly different from zero. The result from the station analysis of a negative trend is well within the uncertainty
range. The re-analysis is compatible with this, but the CPC analysis, although at a lower resolution, does not have the expected bias towards lower extremes. We have not yet investigated the reason for this.


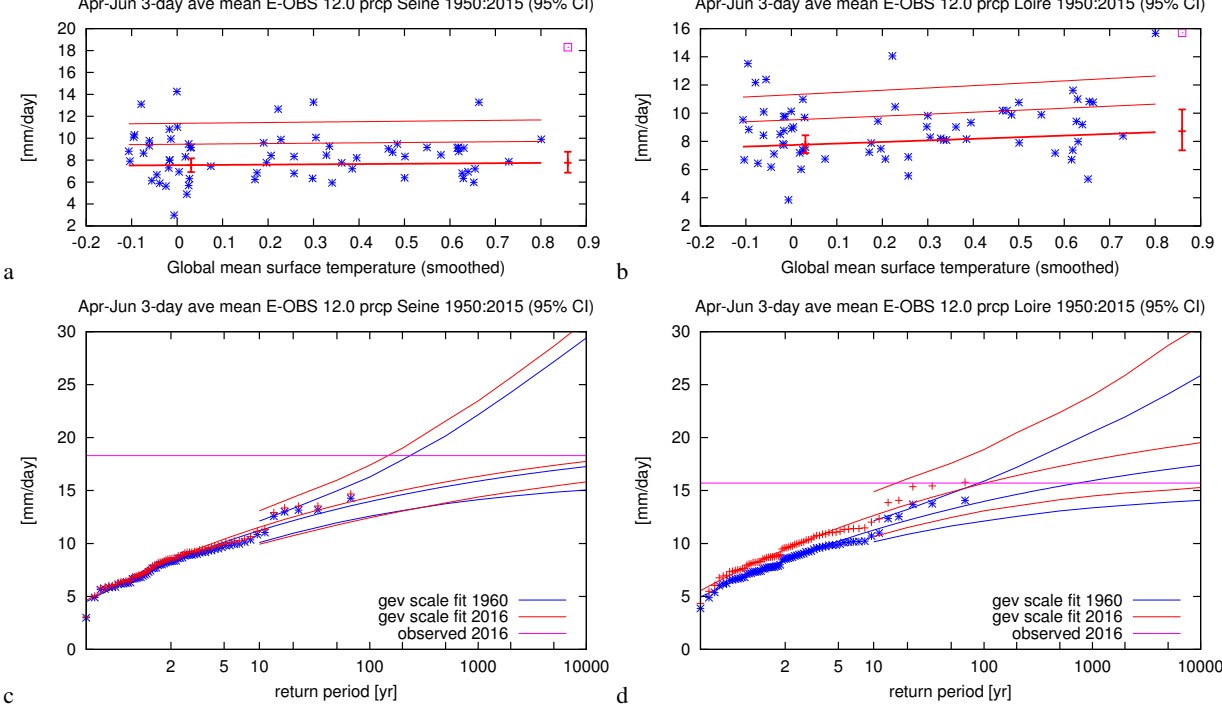

**Figure 4.** Fit of the highest observed 3-day rainfall in the Seine basin in April–June to a GEV that scales with the smoothed global mean temperature. a) The location parameter $\mu$(thick line), $\mu+\sigma$ and $\mu+2\sigma$ (thin lines). The purple square denotes the value of 2016. c) Gumbel plot of the GEV fit in 2016 (red lines) and 1960 (blue lines). The observations are drawn twice, scaled up with the trend to 2016 and scaled down to 1960. b,d) The same for the Loire.

## 4 HadGEM3-A

In the EUropean CLimate Extremes Interpretation and Attribution (EUCLEIA) project, the UK Met Office model HadGEM3-A (Christidis et al., 2013) was run in atmosphere-only mode at high resolution (N216, about 60km) for the period 1960–2013 with observed forcings and sea-surface temperatures (SSTs) ('historical') and with preindustrial forcings and SSTs from which the effect of climate change has been subtracted ('historicalNat'). The latter change has been estimated from the Coupled Model Intercomparison Project phase 5 (CMIP5) ensemble of coupled climate simulations (for details see Stone and Pall, 2016). The simulations were made as an ensemble of 15 realizations for the same forcings. The data are freely available for non-commercial use.

The model maximum 3-day precipitation is about 15% higher than observed in the Seine basin and about 7% in the Loire basin (Table 2). The scale parameters $\sigma$ are overestimated by the same factor, and the shape parameters $\xi$ are compatible with the observed distribution, so that the model results can just be scaled back with a simple bias correction. The extremes in Germany are not represented well, the shape parameter $\xi$ is too large, so that the tail of the distribution is too fat. We therefore





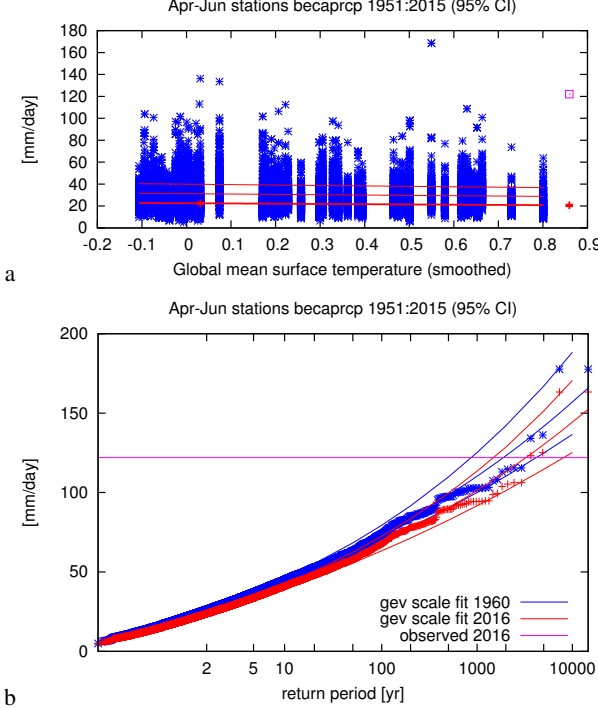

a

b

**Figure 5.** As Fig. 4, but for all stations in 48–51°N, 7–13.5°E (Southern Germany) in the ECA&D dataset with at least 50 years of data.

apply a small bias correction for the Seine and Loire, but do not consider this model for the small-scale thunderstorms in southern Germany.

   Over the Seine basins, the model shows a clear increase in 3-day extreme precipitation in April–June (see Table 2). The return time for an event like the one observed is about 200 yr in the current climate, with a 95% uncertainty range from 100

to 500 yr. This is a factor 1.9 (1.1 to 3.4) more frequent than in 1960, which is significantly different from zero at $p<0.025$. The return time of the observed Loire precipitation is about 40 yr in April–June (23 to 62 yr). In this model the probability increased by a factor 1.2, which is not significantly different from no change.

   Repeating the analysis for the historicalNat we find no trend, RR = 0.95 (0.5 to 2.2) on the Seine and 1.1 (0.5 to 1.7) on the Loire. This shows that the natural forcings and SST patterns have not had a large influence on the trend over this period. The

return time is constant and 370 yr (160 to 1000 yr) for the Seine, 73 yr (50 to 200 yr) for the Loire. Comparing the historical and historicalNat runs we conclude that the probability has increased by a factor 2.0 over the Seine basin due to anthropogenic emissions (0.6 to 7.2). Over the Loire basin the factor is 1.8 (0.7 to 4.1), both of which are not significantly different from no change at $p<0.05$. However, the trend in the historical runs is significantly different from zero and the agreement between the RR from that analysis and from the difference to the historicalNat runs, plus the RR near one in the historicalNat runs, is

evidence that it is mostly due to anthropogenic emissions.





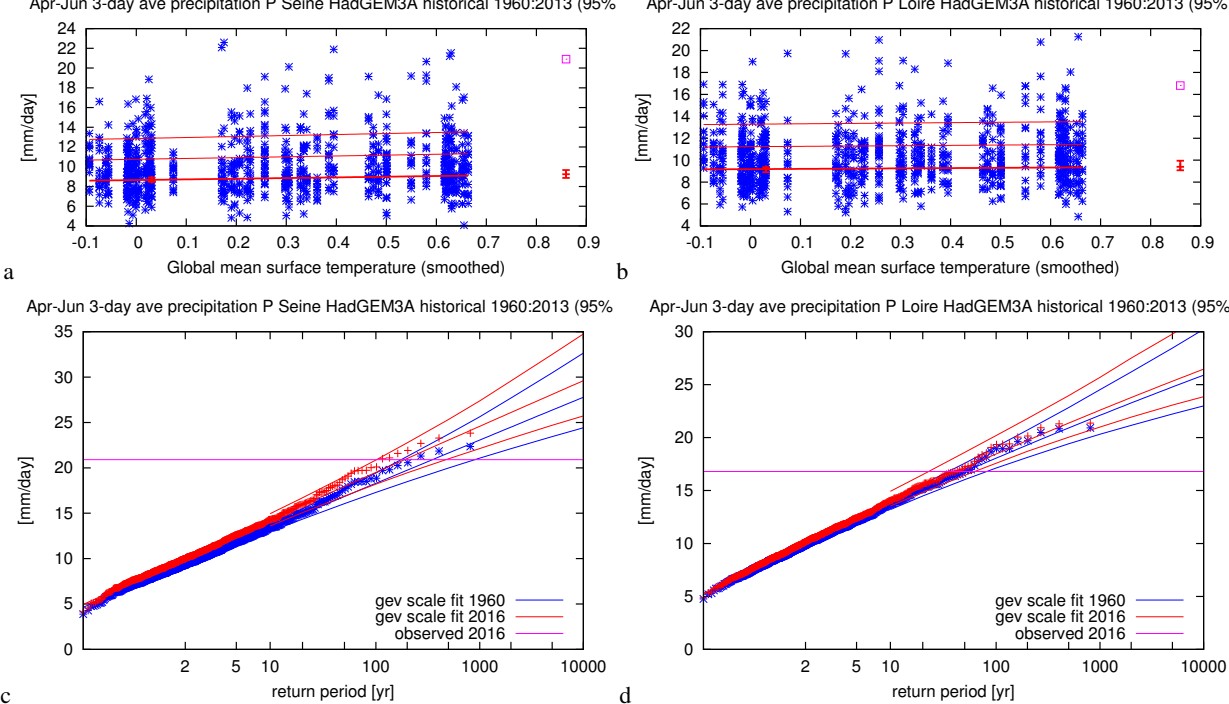

**Figure 6.** As Fig. 4, but for the historical run of the HadGEM3-A N216 model. The 2016 value (horizontal line) has been scaled up to agree with the model bias.

## 5   EC-Earth

We performed the same analysis as in the observations on the coupled general circulation model EC-Earth 2.3 (Hazeleger et al., 2010). We used 16 experiments covering 1861-2100 of this EC-Earth model using the CMIP5 protocol (Taylor et al., 2011). The resolution of the model is T159, this is about 150 km. This relatively coarse resolution means that the Seine and Loire river
5   basins are only six and nine grid boxes respectively.

The EC-Earth runs for the Seine and Loire strongly overestimate the mean precipitation and underestimate the skewness, and therefore also overestimate the position parameter $\mu$ in the GEV fit. As the scale parameter $\sigma$ is on the low side, the distribution cannot be scaled to compensate for this bias (see Tables 2–4 for the parameters) . The results for Germany have the same problem, probably caused by the coarse resolution compared to the phenomena we are investigating. For this reason we do not
10   trust the model results for our three specific domains and time scales. We therefore did not use the EC-Earth model for these events.




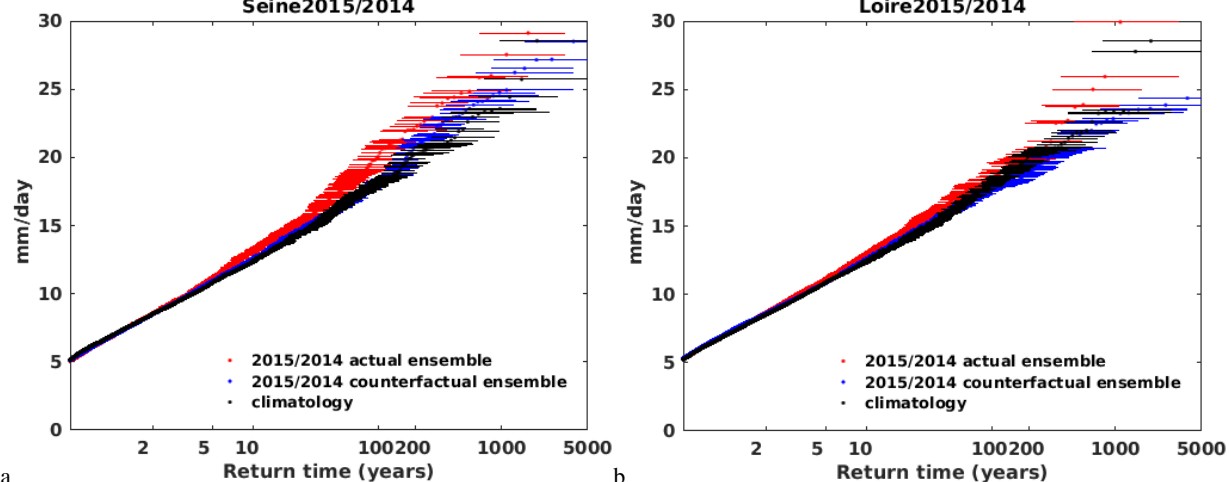

**Figure 7.** Return times for 3-day averaged precipitation in April–June 2014 and 2015 from HadRM3P-EU for the Seine (a) and Loire (b) river basin. Red dots are return times for current conditions ('historical'/ACT) with 95% confidence intervals, blue for counterfactual conditions ('historicalNat'/NAT), and black denotes climatological conditions (CLIM) for the 1986-2014 baseline period. No bias correction has been applied.

## 6 Weather@Home

We use the HadRM3P regional model at 50 km resolution over Europe (Massey et al., 2015). Three different experiments are used for the analysis: (1) Climatology for the period 1986–2014 with observed forcings and observed SSTs ('historical-Clim'/CLIM), (2) Actual experiment with observed forcings and observed SSTs for 2014, 2015 and 2016 ('historical'/ACT) and (3) Natural experiment with preindustrial forcings and counterfactual SSTs for 2014, 2015 and 2016 ('historicalNat'/NAT), obtained by subtracting various estimates of the difference between preindustrial and present-day conditions from CMIP5 (Schaller et al., 2014). The ensemble size is 200 member per year in the CLIM experiments, 1100/2200/200 for 2014/2015/2016 in ACT and 3000/4700/200 in NAT. April, May and June 2014 as well as 2015 are used as daily precipitation for 2016 was not yet available. This assumes that the probability of high precipitation does not depend on SST patterns that are different in 2016 than they were in 2014 and 2015. The 2016 data is used to diagnose the potential dynamic contribution that current SSTs might have had on the European floods.

The availability of large ensembles has always been the advantage of the weather@home approach to attribution. This make it possible to estimate changes in return times without making assumptions to the shape of the distribution or how it depends on the anthropogenic forcing (the assumptions underlying Eq. 1). We also do not use a bias correction for the same reasons, although we note that the 3-day running mean maximum in April–June is underestimated by HadRM3P-EU. As the 1-day extremes are realistic, we believe that this is a model artefact related to insufficient persistence. Rather than bias correcting, we use the best estimate for the observed return time to determine the mean Risk Ratio.





Over the Seine river basin, HadRM3P shows an increase in 3-day accumulated extreme precipitation from the experiments for 2014–2015 with only natural forcings (NAT) to the experiments with all forcings (ACT, see Fig. 7). Comparing the two experiments at the best estimate of the return time from observations, 370 yr, we conclude that the probability of an extreme rainfall event such as observed occurring has increased by a factor 2.0 over the Seine basin (0.6 to 5; Fig. 7 a, not significantly different from no change). Over the Loire river basin, we find a Risk Ratio of 1.8 (1.2 to 2.7; Fig. 7b, significant at $p<0.05$ based on the 73 yr return time. Note that the similarity of the curves in Fig. 7 and in the other figures justifies the assumptions made in the other analyses.

Based on the results with May 2016 SSTs (GloSea5 forecast SSTs from March applied according to the method described in Haustein et al. (2016)), we found no significant contribution of anomalous SSTs patterns to circulation anomalies over Europe. However, the Climatology experiment (black dots in Fig. 7) suggests a strong role for case-specific dynamic contributions in case of the Seine event, but less so for the Loire event. We did not investigate where this originates, but note two known effects. First, long-term observations show on average a wetter spring after El Niño (van Oldenborgh et al., 2000) and Seine run-off. The positive SST in the ENSO region during spring 2015 could have contributed to the offset between current SST conditions and climatology. Secondly, North Atlantic SSTs have been connected to the decadal variability in French river runoff (Boé and Habets, 2014). These were similar in all three years, which therefore should not give different answers for 2016. The higher values for the conditions in 2014–2015 thus agree both with the developing El Niño in 2015 and with the connection with North Atlantic SSTs found in Boé and Habets (2014), both effects increasing the likelihood of wet extremes over climatology in 2015 and even more so 2016.

For Germany, the ability to reproduce observed PDFs of convective precipitation is very limited in our model due to its relatively coarse resolution (50 km) compared to the scale of the thunderstorms. We therefore did not use this model for the German rainfall.

# 7 RACMO

We use a regional downscaling of the 16 member EC-Earth 2.3 ensemble over Western Europe using the RCM KNMI-RACMO2 (Van Meijgaard et al., 2008; van Meijgaard et al., 2012). RACMO2 is run 16 times at a resolution of 0.11°( 12km) — similar to the CORDEX ensemble — for the period 1950–2100, using initial conditions and boundaries derived from the corresponding EC-Earth members. The RACMO2/EC-Earth ensemble members only differ due to internal variability. Thus, for the period 1950-2015 the ensemble provides $16\times66$ years of high resolution data. Although this model cannot yet resolve thunderstorms it is expected that due to the relatively high resolution it reproduces small-scale extremes better than the ≈150 km EC-Earth ensemble, the ≈60 km HadGEM3-A ensemble and the 50 km HadRM3P Weather@Home ensemble (Prein et al., 2015).

The GEV fit parameters for the Seine are compatible with the fit to the observations (Table 2,3), so we accept the model for this analysis and do not apply a bias correction. In the Loire region the model results are about 5% lower than the observed





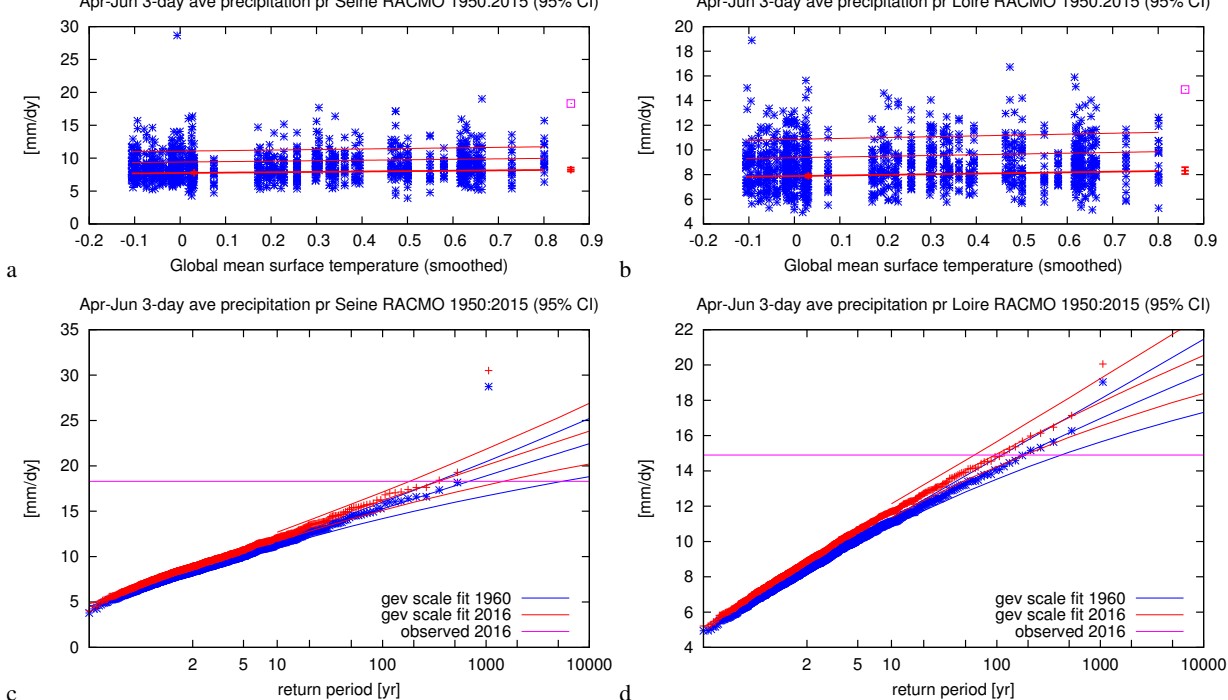

**Figure 8.** As Fig. 4 but for the RACMO data. The observed value (horizontal line) for the Loire has been scaled down by 5% to compensate for the model bias.

ones so we do apply this small bias correction. The distribution in Germany is also similar to the observed one albeit somewhat wetter, so we accept this model there as well with a 15% bias correction.

The RACMO ensemble gives a return time in the current climate of 350 yr (180 to 1500 yr). The probability of an event like the observed one increases with a factor 2.0 (1.3 to 4.9), which is different from no change at $p<0.01$. On the Loire, the return
5   time is about 100 yr (60 to 180 yr) and the RR is 1.8 (1.3 to 3.2), again very significantly different from no change.

The highest 1-day precipitation in southern Germany as observed has in this model a return time of 270 yr (140 to 590 yr). The probability has increased by a RR of 1.7 (1.1 to 2.7) in this model.

# 8   CORDEX

The EURO-CORDEX ensemble (Jacob et al., 2014) was designed to provide a coordinated set of climate projections at a
10   relatively high resolution (12 km), over Europe and part of the North Atlantic. We used a subset of the EURO-CORDEX climate projections. The simulations do not include natural experiments, but all include the historical period starting from dates between 1950 and 1971. We used 8 runs using historical forcings up to 2005 and the RCP8.5 scenario 2006–2015: CNRM-CM5 r1i1p1 / SMHI-RCA4 (1970–2015), EC-EARTH r12i1p1 / SMHI-RCA4 (1970–2015), EC-EARTH r1i1p1 / KNMI-





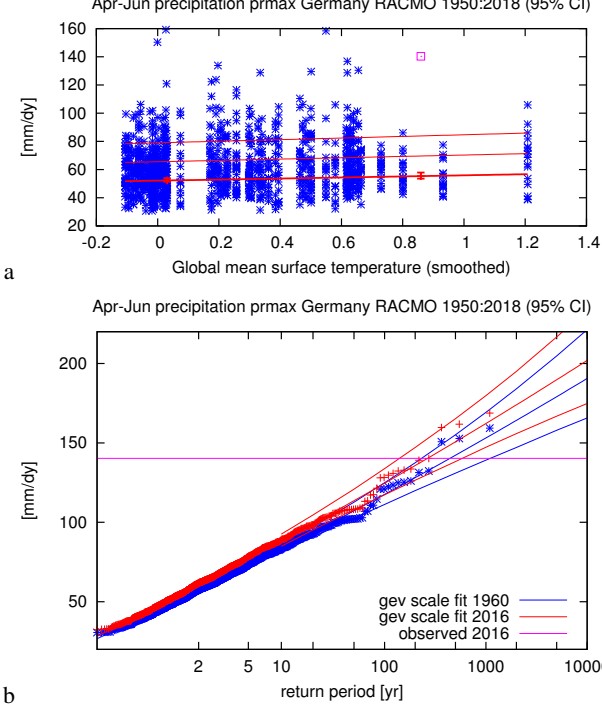

**Figure 9.** Trend fit for the spatial maximum of 1-day precipitation in southern Germany (48–51°N, 7–13°E), using the RACMO ensemble. The observed value (horizontal line) has been scaled up by 15% to compensate for the model bias.

RACMO22E (1950–2015), EC-EARTH r3i1p1 / DMI-HIRHAM5 (1951–2015), IPSL-CM5A-MR r1i1p1 / IPSL-INERIS-WRF331F (1951–2015), IPSL-CM5A-MR r1i1p1 / SMHI-RCA4 (1970–2015), HadGEM2-ES r1i1p1 / SMHI-RCA4 (1971–2015), MPI-ESM-LR r1i1p1 / MPI-CSC-REMO2009 (1950–2015)and MPI-ESM-LR r1i1p1 / SMHI-RCA4 (1971–2015). These all have different biases, these are corrected for to first order by a simple scaling to a common mean. Note that the
5   common driving GCMs and RCMs imply that these experiments are not all independent. The same moving block procedure employed for the station data found that there are only 4 to 5 degrees of freedom in these 8 ensemble members. The uncertainties take this into account.

    The basin averages over the Seine and Loire have a shape that resembles the observed distribution well enough after the multiplicative bias correction. For the Seine we find a return time of about 1000 yr (>250 yr) with a risk ratio of 1.6 (0.5 to
10   4.9). This is not significantly different from no change on its own, but also consistent with the significant changes detected by the other models. For the Loire basin the return time is around 60 yr (36 to 130 yr) with a risk ratio of 1.9 (1.1 to 3.6), which is significantly different from no change at $p<0.05$.

    We did not manage to process the CORDEX simulations for Germany in the near real-time window for the study.




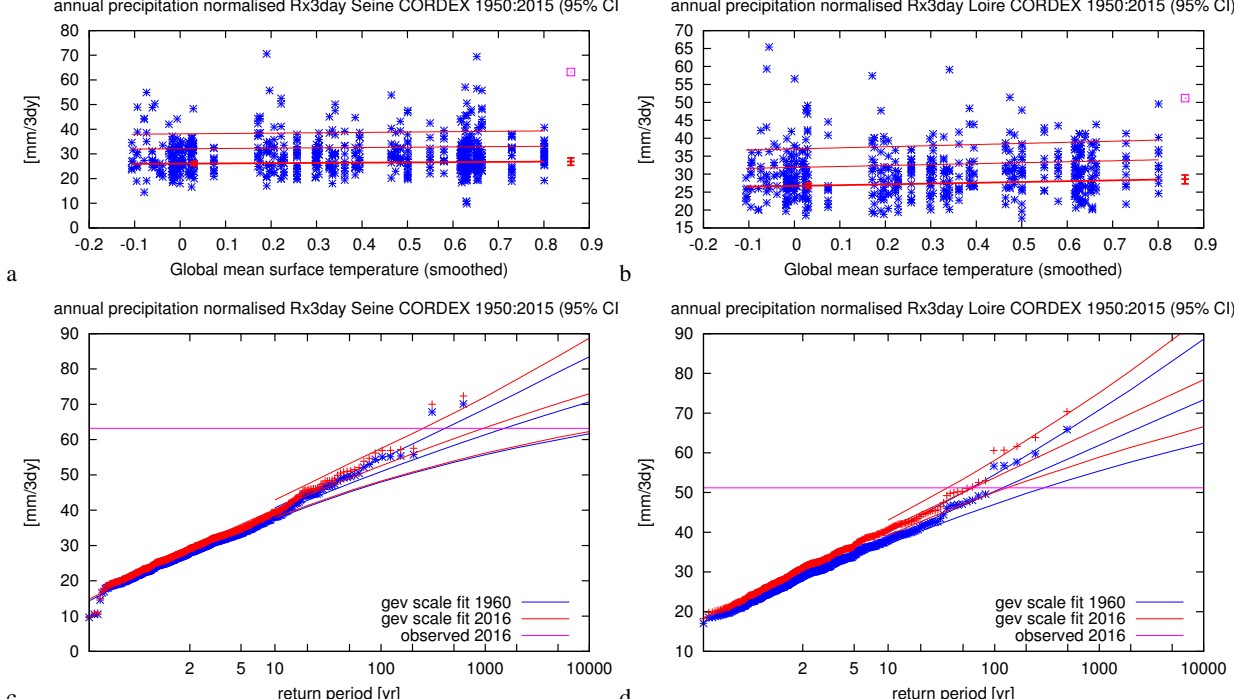

**Figure 10.** As Fig. 4 but for 3-day sums in the CORDEX data.

## 9 Conclusions

Floods on the Seine are rare this time of year, they generally occur in winter. Although the overall return time of the flood crest at Paris was about 20 years, only two late spring/summer floods have been recorded before in over 500 years. We computed that the return time of the 3-day rainfall that was one of the main factors that caused flooding in April-May-June was a few

hundred years in the Seine basin. The Loire has a more complicated seasonality, the maximum 3-day sum of precipitation averaged over the Loire basin in April–June 2016 has a return time of roughly 50 years.

The observational records are too short to establish a trend over the last 65 years. We considered five ensembles of climate model experiments to address the question whether the probability of these kind of events has changed due to anthropogenic emissions of greenhouse gases and aerosols. One relatively coarse-resolution model was not realistic enough for this analysis.

To compare the risk ratios from the trend analyses with the difference between historicalNat and historical conditions we need to convert the 1960–now increase to an estimate of the increase due to anthropogenic forcings. This is done by raising the risk ratios to the power of 3/2, based on the warming up to 1960 being about 1/3 of the total warming.

The summary of the risk ratios is given in Table 1. The uncertainty margins on the different model results only include natural variability. For the Seine, the model spread is well within his range. We just average the five results and add the uncertainties

in quadrature to obtain the combined result. For the Loire, the model spread seems larger than natural variability indicates, but



| Seine | 1960–2016 | natural–2016 |
|---|---|---|
| HadGEM3A | 1.9 (1.1…3.4) | 2.5 (1.1…6.3) |
| HadGEM3A Nat | | 2.0 (0.6…7.2) |
| Weather@Home | | 2.1 (0.6…5.0) |
| RACMO | 2.0 (1.3…4.9) | 2.8 (1.4…11) |
| CORDEX | 1.6 (0.5 to 4.9) | 2.0 (0.3…11) |
| Combined | | 2.3 (>1.6) |
| Loire | 1960–2016 | preind.–2016 |
| HadGEM3A | 1.2 (0.8…2.4) | 1.3 (0.8…3.7) |
| HadGEM3A Nat | | 1.8 (0.7…4.1) |
| Weather@Home | | 1.8 (1.2…2.7) |
| RACMO | 1.8 (1.3…3.2) | 2.5 (1.4…5.8) |
| CORDEX | 2.0 (1.1…3.6) | 2.6 (1.1…6.8) |
| Combined | | 2.0 (>1.4) |
| Germany | 1960–2016 | preind.–2016 |
| ECA&D | 0.6 (0.4…0.7) | 0.4 (0.3…0.6) |
| RACMO | 1.7 (1.1…2.7) | 2.1 (1.1…4.3) |
| No combination possible | | |

**Table 1.** Summary of the risk ratios found with the different methods

is still too small to determine from the inter-model variability. We followed the same procedure but inflated the uncertainty by 30% to account for the model spread.

We can attribute this to anthropogenic climate change using two lines of argument. Firstly, this increase in likelihood is equivalent to an increase in intensity of 6% to 7% for a constant return time. This value is in line with the increase in water

vapour expected due to the Clausius-Clapeyron relation under a constant relative humidity and a heating of slightly under one degree of the Mediterranean and subtropical Atlantic Ocean that are the likely sources of the moisture here, so the trend in precipitation is related to the trend in SSTs that has been attributed to anthropogenic emissions. More directly, the two analyses that explicitly investigate the difference between natural and natural+anthropogenic forcings give results that are compatible with all the others, and together are significantly different from no change, whereas the natural-only runs shows no trend.

Together this confirms that the trend is to a large extent due to anthropogenic forcings.

An unanswered question at the moment is whether a change in the dynamics played a role: have the odds of a stationary cut-off low over this region increased? The methods to answer this question have not yet been developed enough to answer it in the rapid 10-day time window.

The results for Germany are not as clear-cut. A daily precipitation amount as high as the highest value observed is very rare

at an individual station, with a return time of the order of 3000 yr in the current climate from station data. However, because





| Seine | $\mu$ | $\sigma$ | $\xi$ | $\tau_1$ | RR |
|---|---|---|---|---|---|
| dataset | mm/dy | mm/dy | | year | |
| CPC | 7.1 (6.7…7.6) | 1.8 (0.9…1.5) | -0.35 (-0.14…0.11) | | |
| E-OBS | 7.8 (7.2…8.2) | 2.0 (1.4…2.3) | -0.15 (-0.21…0.10) | 31000 (>150) | 4 (>0.001) |
| ERA-interim | 7.6 (6.9…8.2) | 1.7 (1.2…2.0) | -0.12 (-0.17…0.10) | | |
| HadGEM3A | 9.2 (9.1…9.4) | 2.2 (2.1…2.4) | -0.004 (-0.049…0.047) | 190 (100…500) | 1.9 (1.1…3.4) |
| HadGEM3A Nat | 8.8 (8.6…9.0) | 2.4 (2.2…2.5) | -0.049 (-0.093…0.000) | 370 (160…1000) | 0.95 (0.47…2.2) |
| EC-Earth | 9.0 (8.9…9.1) | 1.7 (1.7…1.8) | -0.018 (-0.050…0.006) | – | – |
| weather@home | 7.6 (7.5…7.7) | 2.2 (2.1…2.3) | -0.03 (-0.04…-0.015) | – | 2.1 (0.6…5) |
| RACMO | 8.2 (8.1…8.4) | 1.8 (1.6…1.8) | -0.010 (-0.075…0.028) | 350 (180…1500) | 2.0 (1.3…4.9) |
| CORDEX | 9.0 (8.8…9.2) | 2.1 (1.8…2.3) | -0.05 (-0.11…0.03) | 960 (250…15000) | 1.6 (0.5…4.9) |

**Table 2.** Summary of the fits of the data for the Seine: location parameter $\mu$, the scale parameter $\sigma$ and the shape parameter $\xi$ of the GEV fit for 2016, return time of the event in 2016 and risk ratio relative to 1960 (1979 for CPC and ERA-interim). Values between brackets denote the 95% interval.

there are almost 300 stations the probability of hitting any of these with this high precipitation is much higher. The return time of daily precipitation this high somewhere in southern Germany is of the order of 20 yr.

A pooled analysis of all stations with 50 years or more of data points to a *decrease* in the probability of very high 1-day precipitation events with a factor of 0.6 (0.4 to 0.7) since 1960. This factor is very similar when the series are normalised to the same mean first. This decrease is surprising, as similar fits to observations in the Netherlands (unpublished), southern France (Vautard et al., 2015) and Jakarta (Siswanto et al., 2015) showed clear increases. A possible cause would be lack of moisture availability as this region is much further from the oceans than the three examples above.

There is only one model (RACMO) that seems to represent the statistical properties of the heavy thunderstorms that caused most problems correctly, although convection is still parametrised in this model. This model shows an increase in likelihood with a factor 1.7 (1.1 to 2.7), which is not compatible with the observed change. We cannot ascertain at his moment which of the two analyses is more realistic, so we cannot make a statement on the trend of heavy 1-day precipitation in southern Germany.

*Acknowledgements.* We thank Peter Uhe and Julie Arrighi for a careful and critical reading of the manuscript. The Deutsche Wetterdienst (DWD), ECA&D and NOAA/CPC are acknowledged for providing the historical observational data and ECMWF for the (re)analyses. For the climate model ensemble data we thank the Met Office for the provision of the HadGEM3-A data; all of the volunteers of weather@home who have donated their computing time to generate the large ensemble simulations; all of the participants who computed simulations for climateprediction.net; our colleagues at the Oxford eResearch Centre: A. Bowery, M. Rashid, S. Sparrow and D. Wallom for their technical expertise; the Met Office Hadley Centre PRECIS team for their technical and scientific support for the development and application of weather@home; our colleagues Camiel Severijns and Erik van Meijgaard at KNMI for their efforts in producing the 16-member EC-Earth and RACMO/EC-Earth ensembles; and the coordination and participating institutes of the EURO-CORDEX initiative for producing and





| Loire | $\mu$ | $\sigma$ | $\xi$ | $\tau_1$ | RR |
|---|---|---|---|---|---|
| dataset | mm/dy | mm/dy | | year | |
| CPC | 8.9 (8.2…9.7) | 2.4 (1.6…2.8) | -0.14 (-0.46…0.04) | | |
| E-OBS | 8.7 (8.3…9.2) | 2.0 (1.4…2.4) | -0.13 (-0.22…0.09) | 100 (>15) | 7 (>0.1) |
| ERA-interim | 9.0 (8.3…9.8) | 1.9 (1.2…2.3) | -0.05 (-0.30…0.22) | | |
| HadGEM3A | 9.4 (9.2…9.5) | 2.1 (2.0…2.2) | -0.027 (-0.074…0.026) | 42 (23…62) | 1.2 (0.8…2.4) |
| HadGEM3A Nat | 9.3 (9.1…9.5) | 2.0 (1.8…2.1) | -0.06 (-0.11…-0.02) | 73 (50…200) | 1.1 (0.5…1.7) |
| EC-Earth | 14.2 (14.0…14.3) | 2.8 (2.6…2.8) | -0.04 (-0.07…-0.01) | – | – |
| weather@home | 7.8 (7.7…7.9) | 2.2 (2.1…2.3) | -0.035 (-0.045…-0.015) | – | 1.8 (1.2…2.7) |
| RACMO | 8.3 (8.2…8.4) | 1.6 (1.5…1.7) | -0.04 (-0.09…0.00 | 96 (61…180) | 1.8 (1.3…3.2) |
| CORDEX | 9.5 (9.3…9.7) | 1.8 (1.7…2.0) | -0.01 (-0.07…0.05) | 62 (36…130) | 1.9 (1.1…3.6) |

**Table 3.** Summary of the fits of the data for the Loire: location parameter $\mu$, the scale parameter $\sigma$ and the shape parameter $\xi$ of the GEV fit in 2016, return time of the event in 2016 and risk ratio relative to 1960 (1979 for CPC and ERA-interim). Values between brackets denote the 95% interval.

| Germany | $\mu$ | $\sigma$ | $\xi$ | $\tau_1$ | RR |
|---|---|---|---|---|---|
| dataset | mm/dy | mm/dy | | year | |
| ECA&D | 68 (63…74) | 17 (14…20) | 0.02 (-0.21…0.18) | 23 (8…1800) | 0.8 (0.014…2.7) |
| E-OBS | 47 (44…52) | 12 (9.6…14) | -0.03 (-0.19…0.11) | 32 (9…200) | 1.4 (0.24…11) |
| CPC | 70 (65…75) | 16 (10…19) | 0.13 (-0.30…0.06) | | |
| ERA-interim | 46 (25…31) | 10 (7…12) | 0.05 (-0.12…0.23) | | |
| HadGEM3A | 57 (56…59) | 16 (15…17) | 0.16 (0.12…0.24) | – | – |
| EC-Earth | 24 (23…24) | 5.3 (5.0…5.4) | 0.04 (0.01…0.07) | | |
| weather@home | 55 (54…46) | 15 (14…16) | -0.02 (-0.025…-0.015) | – | – |
| RACMO | 55 (55…57) | 14 (13…15) | 0.02 (-0.02…0.07) | 270 (140…590) | 1.7 (1.1…2.7) |

**Table 4.** Summary of the fits of the data for Germany: location parameter $\mu$, the scale parameter $\sigma$ and the shape parameter $\xi$ of the GEV fit in 2016, return time of the event in 2016 and risk ratio relative to 1960 (1979 for CPC and ERA-interim). Values between brackets denote the 95% interval.



providing their model output. This project was supported by the World Weather Attribution initiative and the EU project EUCLEIA under Grant Agreement 607085.



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
