# Peer review of "Rapid attribution of the May/June 2016 flood-inducing precipitation in France and Germany to climate change"

_Hydrology and Earth System Sciences, 2016_

## Referee Comment (RC1) · Anonymous Referee #1 · 5 Aug 2016

**Manuscript summary**

This manuscript proposes a rapid attribution to anthropogenic climate change of a type of precipitation event – location: Seine and Loire catchments, southern Germany; season: April-June; intensity: 3-day maximum/1-day maximum – that actually occurred in May-June 2016. This actual precipitation event notably led (together with wet antecedent conditions) to floodings in France and Germany. The authors make use of a range of GCM/RCM runs as well as observational data to compute the ratio of the probability of occurrence of this type of event (1) between 1960 and 2016 in the actual world, and (2) in 2016 (or 2014/2015) in the actual world or in a counterfactual

world with preindustrial greenhouse gas concentrations and sea surface temperature with the effect of anthropogenic climate change removed. They found an increase in the probability of occurrence in both cases for the Seine and Loire catchments, but not significant in all experiments. Results for Germany appear less robust.

**General comments**

My first general comment is that the issue of flood event attribution to anthropogenic climate change is scientifically sound and within the scope of HESS. However, the quality of this "rapid attribution *study*" is questionable due to constraints imposed by the short time frame, and the manuscript proposes a "quick-and-dirty" attribution *report* of the study. The reasons for providing this rather strong statements are developed below:

1. First, let me ask the following question: what is the purpose of this "rapid attribution *study*"? (please note that I am not yet referring to the manuscript) The authors claim that it is motivated by demand on such information: "The extreme nature of this event left many asking whether..." (P1L7), "However demand on such information is often in real time, when, for a couple of days, damages and losses raise the attention to the public and media." (P5L3-4). Consequently, as put by the authors, "A challenge is therefore to provide scientifically sound and reliable information in near real time (about a week) about human influence on extreme events" (P5L4-5). The question here is: who expresses this demand? Is it the general public and the media, as suggested by the authors? If so, what is the actual societal use of delivering such information on such a short time? One may argue that it contributes to the public awareness of the local consequences of anthropogenic climate change by resonating with the short-term memory. I personally find it a weak argument, and I believe that it is no sufficient to drive such a "rapid" study. Conversely, I would perfectly understand performing an attribution study – without the "rapid", as I will detail below – as a way for government, local authorities, or regulators to inform climate change adaptation strategies.

2. Coming now to the manuscript. It relates this "rapid attribution study", results of which "were completed and released to the public in one week" (abstract, P1L10). The authors have to be congratulated for this performance. However, do scientists really have to be congratulated for performing a study with such a speed? Would the study have been "quick-and-clean", the answer would be definitely yes. But according to the authors themselves, the short time frame imposed severe constraints:

   • "However, the data required to analyse the event at the sub-daily scale in real time is not yet available to us (although it is publicly available at DWD)." (P3L31-32)

   • "As French precipitation data were not available in real time, the analyses there are based on a relatively sparse subset of stations." (P7L12-13)

   • "These values were taken to represent the observed event in the following as the E-OBS data for 29-31 May 2016 was not yet available." (P717-18)

   • "We have not yet investigated the reason for this." (P8L16)

   • "We did not managed to process the CORDEX simulations for Germany in the near real time window for the study." (P15L13)

   • "The methods to answer this question have not yet been developed enough to answer it in the rapid 10-day time window." (P17L12-13)

The main issue here is the observational data used. The authors relied on data available in real-time (namely gridded products based on a sparse network of stations), i.e. both sparse and not quality-controlled data. The use of extreme values from such data would at least require checking them against the best available data, which are usually available a month later (for manned rain gauges). The

fact that no radar data was used in the study was also surprising as these are usually available in real time, even if their quality may be discussed (but at least they offer a detailed spatial view of rainfields contributing to a robust estimate of catchment-average precipitation). A secondary issue here concerns the variable used for this attribution study. In France, it focuses on 3-day precipitation, and not on the actual streamflow values reached during the flooding event. The authors are aware that factors other than this high precipitation intensity came into play:

- "[...] many other contributing factors are neglected in this rapid attribution study." (P3L15)
- "Firstly the soil types and saturation levels at the time of this extreme rainfall event have not been captured." (P3L15-16)
- "This analysis also does not take into account the impacts of the reservoirs [...]" (P3L17-18)
- "In addition, land cover and associated runoff characteristics have also not been taken into account" (P3L19)
- "A full attribution of the flood themselves, rather than just the rainfall event, would need to take all of these factors into account." (P3L19-20)

It appears all the more disappointing from the hydrological point of view that such a streamflow attribution study would have been possible (or at least a subset of the experiments) with the help of catchment hydrologists, had the "imposed" time frame not been so short. As a conclusion to this comment, I would ask this question: is the speed at which a scientific study is performed a positive point for evaluating a corresponding manuscript? Can it compensate other negative consequences of the study resulting from the reduced time frame? I will leave the answer to the editor, but from my point of view, this is clearly no.

3. Coming now to the *contents* of the manuscript. The authors claim that it took them "an additional week to finalise this article" (abstract, P1L10-11). Again, I

believe that the authors should be congratulated provided that their manuscript is clear and sound. However, this is rather not the case:

- The manuscript is definitely not well organized: there is no Data nor Discussion sections. Results and discussions and intertwined. Some discussion-relevant elements also appear in the introduction. Results are commented in the conclusions, etc.

- The central method for deriving risk ratios between now and earlier in the 20th century, and for comparing them with risk ratios derived from fac-tual/counterfactual worlds is not justified nor detailed enough.

- The text is vague in many locations, on methods, on the use of GCM/RCM data, on the interpretation of results, etc. It therefore makes the study not reproducible at all.

- Several figures are not referred to in the text.

- There are several inconsistencies between the text and figures/tables.

- etc.

This manuscript therefore clearly makes the reader feels it has been written in a hurry (like in one week), while it also suggests that the scientific underlying content may be rather valuable (with the restrictions mentioned above). Trying never-theless to adopt a constructive approach, I took approximately half the time of the authors' writing to identify and list all the points that could/should be improved in the manuscript. This (long) list is given in the Detailed comments section below.

In conclusion, I would recommend the editor to reject the manuscript, and to invite the authors to resubmit a manuscript to HESS. I would recommend this new manuscript to be written without the – presumably – artificial time constraints, and to be based on higher-quality observational data that has been made available since their initial

submission. I would welcome any further discussion with the authors, the editor, and any other scientific contributor, as I believe the issues raised above are of general significance as they allude to possible drifting ethics in science and science publishing.

**Detailed comments**

Abstract

1. P1L18 "all four climate models": what are they? Please define clearly the attribution set-up.

2. P1L17 "has increased": between what and what?

3. P1L19 "The observed trend": over what period?

4. Figure 1, a and b: It would be great to have the underlying network of precipitation stations. This should be at least available for E-OBS. And please add a scale to each map.

Introduction

5. P2L13 "most severe event": in terms of what?

6. P2L14 "25% of the flood peak of the Seine": How is this estimated? Please provide a reference or a method.

7. P2L15 "height above 6.1 m": At what hydrometric station?

8. P2L16-17 "Some measurement problems...": What kind of problems? Please clarify this.

9. P2L19 "of about 20 yr": How is it estimated? Any reference?

10. P2L28 "forced to close": on what day?

11. P2L28 "without electricity": where?

12. P3L2-3 "46% above the previous records": Reference?

13. P3L9 "$-40m^3.s^{-1}$": for a peak flow of...? As such, this figure is not informative.

14. P3L10 "3-day precipitation": Well, this France-averaged estimation is not relevant for all basins. It first heavily depends on the catchment size, but also on flood-generation processes which are catchment-specific. The relevant precipitation time scale for catchments located in the Cévennes area (south-eastern fringe of the Massif Central) is much closer to a few hours, whereas it is several (and usually more than three) days for the Seine@Paris due to the buffering effects of large aquifers. Please better justify your choices here, as the whole study depends on it.

15. Figure 2a: What is represented with light blue and orange colours? Please add a legend, and a scale.

16. Figure 2b. Please add a scale and remove the title.

17. Figure 2: The reader should be able to compare the model grid scales and limits (see "which ends at 13$^e$E" in the legend) to the observation network and density. Please add such grid scales in some way to this figure.

18. P3L11 "close to the response time": how is it estimated? And on what rivers? With what catchment size? Cf. also above comment.

19. P3L31-32 "However, the data required to analyse the event at the sub-daily scale in real time is not yet available to us (although it is publicly available at DWD)": I don't understand. If they are publicly available, why did you not use them?

20. P5L14 to P6L8: In my point of view, these paragraphs related to future projections are not relevant in the introduction. Results on observed trends and anthropogenic attribution may however be qualitatively checked against findings for 21st century projections in the discussion.
21. P6L9-14: Please summarise the attribution set-up in a few words. The reader may not understand what kind of models are referred to P6L13.

Methods

22. Section 2: The whole "Methods" section is quite unclear. Below are (some of the many) points that need clarification, additional references, etc.

   (a) P6L19 "4-yr smoothed global mean temperature": Please justify the use of the global temperature as an indicator for anthropogenic climate change. For example, why wouldn't you alternatively use the local/regional temperature like in Vautard et al. (2015)? Or maybe the CO2-equivalent greenhouse gas concentrations? This would more in phase with the GCM counterfactual set-up. Less importantly, please also justify the use of a 4-yr smoothing.

   (b) Equation (1): Please define $T'$. Is it the global mean temperature?

   (c) P6L25 "values larger than about 0.4 are penalised as unphysical": Please give a reference for documenting and justifying this penalised approach.

   (d) P6L26-27 "but take correlations [...] with a moving block technique": Please detail and clarify.

   (e) P6L30 "We evaluate these for the year 2016": I presume this means that you evaluate the cumulative probability density with the temperature of 2016. And so what is this temperature, given that annual temperature for year 2016 is not available? Spring temperature? Please clarify. Same for P7L2-3.

   (f) P7L1: This is actually not a trend detection. This is only a ratio of probability in two specific years, without any formal statistical test, and therefore there is no trend, and no detection. Please rephrase.

   (g) P7L2: Please justify the use of year 1960. Results might have been very different with year 1940 when global temperature first peaked. See also P7L8-9 "the change from 1960 to now...".
23. Section 2: Reading through this section highlights the fact that a "Data" section is definitely missing before it. Indeed, one cannot grasp the meaning of "neighbouring stations or ensemble members" (P6L27), "(analysis and reanalysis)" (P6L29), "Models" (P6L29), "observed record and reliable models" (P7L1), "two models that also have experiments..." (P7L6-7)

24. P7L8-9 "Often we can neglect the effect of natural forcings on these extremes": What does this mean? Please clarify.

Observational analysis

25. P7L12-13: Again, there should be a map of stations used in the gridded products considered here.

26. P7L14-15 "We checked [...] for the past": I don't understand. What is it about decorrelation scales? Why mentioning ERA-Interim here? Is is used at all?

27. P7L15 "Satellite-derived ...": I presume this sentence attempts to justifies the fact that such products are not used here? Please make it explicit and provide some references for their possible low quality of satellite-derived products. And what about radar data? Such products are available in real-time, aren't they? Please provide at least a comment on that.

28. P7L16: How is the CPC gridded product derived from gauges? Please provide some references, and the grid definition. And what about the temporal homogeneity of the underlying network of stations? I mean, is the list of stations used the same in 1960 than in 2016? This is a critical part of the analysis.

29. P7L17, Fig. 3: Please use bar charts for plotting precipitation amounts. Plus, what is the smoother line? Climatology? Over what period? Please add a legend. And remove unnecessary text from the figure titles.

[Figure]

30. P7L21 "The trend analysis": There is no trend analysis in Eq. 1. Please clarify.

31. P7L21-22 "larger than can be determined with the longest series": First, what does this mean? Second, what is the longest series?

32. P7L23 "The best fit for the trend is positive": Please rephrase.

33. P7L23 "However, the uncertainties are large and easily encompass zero": zero what? Are these uncertainties related to the red vertical bars that appear in (but are not commented nor even mentioned in the legend of) Fig. 4a?.

34. P7L25-26 "We can improve the estimate": What do you mean exactly by "improving"? Please clarify.

35. P7L25-27 and Fig. 4b: This is really confusing. Precipitation values mentioned above (P7L16-17) are a sum over 3 days, the plot title of Fig. 4c) suggests a daily average, the plot presumably shows the daily average, and the legend says "3-day rainfall". I would strongly suggest that all plots are made with 3-day sums, not to confuse the reader with the alternative possibility of studying one-day extremes.

36. P7L25-27 and Fig. 4b: Another highly confusing point: the text mentions that the fit is eventually made with a Gumbel distribution. However, the legend of Fig. 4 says that what is plotted is a GEV fit. Please clarify. What adds again to the confusion is the unsuited use of the term "Gumbel plot" for GEV fits...

37. P7L32-33 "The number [...] recently": This is confusing. What is the number of stations in 1951 for example? Please clarify and rephrase.

38. P8L2-5: This is clearly not enough supported by details on the procedures or relevant references. Please provide some more details/references.

39. P8L6: What is the "present climate"? Please clarify.

40. P8L7-8 "The probability [...] this area": This statement is not supported by any figure, or am I wrong? Please clarify.

41. P8L9-10 "... so the spatial heterogeneity". This is unclear, please rephrase.

42. P8L15 "The re-analysis": What reanalysis? ERA-Interim? I thought you were discussing about E-OBS? Please clarify.

Sections 4 to 8

43. General comment for these sections: model runs should be presented (period, grid, forcings) beforehand in the data section. And Sections 3 to 8 reorganized in a Results section.

44. P9L4: Please specify what are the forcings.

45. P9L7: How do you deal with the ensemble of realizations in the fitting procedure? I presume you put all realizations together in order to reduce the sampling uncertainty. Please clarify.

46. P9L7 "for the same forcings": Please rephrase.

47. P9L7-8 "The data... use.": Wouldn't it be better suited in the acknowledgement section with a contact or website?

48. P9L1-8: What is the spatial resolution of HadGEM3-A? Again, please make a map of the model grid. This is critical to justify the comparison with observed (well, gridded) data.

49. P9L9-10 "The model ... (table 2)": Once again, this is really confusing. Is this statement valid for both forcings? Is it relative to the location parameter $\mu$? If yes, I can't see how you may obtain the figures mentioned in the text from Table 2.

50. P10L3: Table 2 only shows a difference in the probability of occurrence of 55 mm/3dy between 1960 and 2016, not an "increase in extreme precipitation". Please rephrase.

51. P10L5 "at $p < 0.025$": What is the statistical test used? Again P10L13.

52. P10L8-9: Again there is no trend here. Only a difference between estimates for two different years.

53. P10L10-12 "2.0 [...] (0.6 to 7.2)": Please detail how these figures are obtained.

54. P10L13-15: This is unclear. Please rephrase.

55. Figure 6: This figure is not referred to in the text. In the legend the model acronym is not consistent with the text and tables.

56. P11L2 "CMIP5 protocol": Please detail this protocol. I presume this is the *historical* runs until 2005. What about afterwards?

57. P11L6-11: This overall negative assessment is really interesting and useful to the community. Same comment for P13L19-21.

58. P12L3: Please make it clear what is the difference between the first experiment (Climatology) and the other two.

59. P12L10-11 "The 2016 data": I don't understand. Please clarify and detail all data types (variables, etc.) used and their specific purposes for the study.

60. P12L12 "The availability ... attribution." Could you give some examples?

61. P12L13-14 "how it depends ... Eq. 1)": This highlights the lack of comments on that point (noted in an above comment) when introducing Eq. 1.

62. P12L17: Please stick to RR once you defined it. Again P13L4, P13L5.

63. Figure 7: The legend mentions "dots", but plots show also lines. What do they represent? I presume it is an envelop of the individual members, but why not pooling them (as for HadGEM, if I am right)? Please clarify.

64. Figure 7: Please display the observational value for 2016 in the plots.

65. P13L6-7 "Note that ... other analyses.": I don't understand. Please clarify.

66. P13L8: What is GloSea5?

67. P13L8-18: This should belong to a discussion part.

68. P13L9 "We found ... over Europe.": Could you explain?

69. P13L10-11 "However ... event": Again, this is not clearly enough explained.

70. P13L12 "and Seine run-off.": There are presumably missing words in the sentence. If this aims at suggesting that Seine runoff is higher than normal in post-Niño years, please provide a reference.

71. P13L25 "CORDEX": First use of the term. Please define.

72. P13L26 "internal variability": Please specify that this is EC-Earth (and not RACMO2) internal variability.

73. P13L31 to P14L1: Please make sure that bias values are consistent with results shown in Table 2 and Table 3.

74. P14L1-2: Please refer to Table 4.

75. P14L3-5: Please refer to Fig. 8. This Figure is not referred to in the manuscript.

76. P14L4-5: Again, what is the statistical test? What does "very significantly" mean?

77. P14L6-7: Please refer to Fig. 9. This Figure is not referred to in the manuscript.

78. P14L12: Please justify the use of runs forced by RCP8.5. I know this may have very little influence, but I'd like it to be commented.

79. P15L4 "biases": I presume on precipitation?

80. P15L4 "simple scaling to a common mean": This is the second time in the paper that this procedure is refereed to, but it is still unclear what is this common mean. Do you simply divide by the observed mean? In that case, at what spatial resolution? Please clarify.

81. P15L6-7 "The uncertainties take that into account": Well, this is far from being sufficient as an explanation, and far from being reproducible. Please detail.

82. P15L8 "The basin averages over the Seine and Loire": I presume you mean "the distribution of basin-average April-June 3-day maximum precipitation"... Please try and be more accurate.

83. P15L8-12: Please refer to Fig. 10. This Figure is not referred to in the manuscript.

84. P15L10-11 "This is ... other models": Is it a formal statement or a more qualitative one?

Conclusions

85. P16L2 "Floods on the Seine are rare this time of the year" and P16L2 "only two late spring/summer floods have been recorded before in over 500 years": Well, I disagree factually. Out of the 30 remarkable flood events identified for the EU Floods Directive (EU, 2007) in the "Seine-Normandie district", eight occurred during April to June (Lang and Cœur, 2014, p. 386): April-May 1983, 16-17 June

1997, April-May 1998, 7-13 May 2000, March-April 2001, 1 June 2003, 7-8 June 2007, 14 June 2009. Among these only, already two show a very similar pattern of soil saturation followed by intense rain, on areas close or very close to those hit by the 2016 rainfall event: the 10-15 April 1983 flood particularly hit the Essonne subcatchment (Lang and Cœur, 2014, p. 404-405), the April-May 1998 flood hit the Yonne and Loing (mentioned P2L7 for 2016) subcatchments (Lang and Cœur, 2014, p. 415-416). But there is also (for example) the 16-23 March 1978 flood that hit the small tributaries south of Paris, including the Yvette river mentioned P2L7 (Lang and Cœur, 2014, p. 403-404).

86. P16L7 "The observational records are too short to establish a trend over the last 65 years.": I don't understand. Is 65 years too short a period to derive a robust trend? As for the length of observational records, the precipitation series available from Météo-France over the Seine basin allow for a computation of basin-scale daily average as least as reliable of this from E-OBS, and for a much longer period. Please rephrase.

87. P16L1012: This should belong to the Methods section. Plus, this is rather unclear as such.

88. P16L14-15 "We just ... result": Please rephrase and detail.

89. P16L14 to P17L2: Some of this should also belong to the Methods section.

90. Table 1: Why does it appear as Table 1 as it is only commented after the two other ones? Plus, I presume you meant "natural-2016" on row 8 column 4.

91. P17L3-13: This belongs to the Discussion section.

92. P18L5-7: I am not sure that comparing the trends in extreme precipitation values in Germany with that of the Cévennes range (with high orographic effects) and Jakarta (in a tropical setting with monsoon influence) is necessarily relevant...

93. P18L11 "the two analyses": What are they?

**Technical corrections**

1. P1L14: "return time" → "return period", and throughout the whole manuscript. This is the most commonly used terminology in hydrology.

2. P1L16: "once roughly" → "roughly once"

3. P1L17 "Seine a factor": probably a missing word

4. P2L2: "rainfalls" → "rainfall"

5. P2L22: "less" → "lower"

6. P2L22: I believe you mean "March 2016"

7. P2L23: the official name is "EU Sequana 2016"

8. P2L11: "seizes" → "sizes"

9. P7L3 and P8L12: "ration" → "ratio"

10. P7L16: "55,mm" → "55 mm"

11. P12L13: "assumptions to" → "assumptions on"

12. P16L2: "flood crest" → "flood peak"

13. P16L14 "his" → "its"

14. P18L10 "his" → "this"

15. P23L2: "precipitations" → "precipitation"

**References**

EU (2007). Directive 2007/60/EC of the European Parliament and of the Council of 23 October 2007 on the assessment and management of flood risks. *Official Journal of European Communities*, L 288/27.

Lang, M. and Cœur, D., editors (2014). *Les inondations remarquables en France – Inventaire 2011 pour la directive Inondation*. Quae.

Vautard, R., Yiou, P., van Oldenborgh, G.-J., Lenderink, G., Thao, S., Ribes, A., Planton, S., Dubuisson, B., and Soubeyroux, J.-M. (2015). Extreme Fall 2014 precipitation in the Cévennes mountains. *Bulletin of the American Meteorological Society*, 96(12):S56–S60.
* * *

---

## Short Comment (SC1) · 26 Oct 2016

Reviewer 1 has thoroughly identified a number of reasons why this paper is nowhere close to being acceptable as a primary scientific publication, so I won't belabor them but will simply say that I agree:

(i) Authors weren't prepared to wait to get access to quality-controlled data;

(ii) Authors weren't prepared to take the time to explore mechanisms for the change in risk, and in particular whether dynamical changes might have been important (which is ironic, since some of the same authors have recently argued that one must do this in event attribution);

[Figure]

(iii) Although the public concern was about flooding and HESS is a hydrology journal, the authors weren't prepared to take the time to assess the flooding.

This particular paper has to be judged on its own merits. However, similar issues arose in the previous HESS submission from this team on the Desmond Storm (http://www.hydrol-earth-syst-sci-discuss.net/hess-2015-534/). The issues would seem to be endemic to the rapid-attribution framework, because they result from the severe time constraints. Thus I would like to take the opportunity that this open review process provides to raise some broader issues about such studies, picking up on the final sentence in Reviewer 1's report.

I am concerned that such studies are a disservice to the scientific community. Peer review is the foundation of science, but is done on a voluntary basis. As a former journal editor I am acutely aware of the enormous effort provided gratis by editors and reviewers. To ask them to assess papers that are written so hastily, with so many details left unaddressed, is simply not fair. Time constraints imposed by the media are not a sufficient reason to rush the process of preparing a paper that should meet the standards of rigour expected of an original scientific publication.

Perhaps more importantly, I am also concerned that they are a disservice to the public, and to the public perception of science. There are two aspects to this. The first is that the public wants to know how the extreme event (in this case flooding) was affected by climate change. What they get is a quantitative answer to a different question, based on some proxy for the event (in this case, precipitation over a large region). As the authors are fully aware, the event definition has a very large effect on the quantitative answer, especially when the latter is expressed in terms of return time. So whilst there is a quantitative answer, it is not serving any local resilience need because it is not about the event that captured people's attention. I would call this pseudo-quantification.

The second aspect is that the Discussion paper looks like a scientific publication, and I suspect the public are unaware that it is not a peer-reviewed publication. I was asked
by a reporter (a science writer at Associated Press, so mainstream) to comment on the third paper in this series, on the Louisiana flooding (http://www.hydrol-earth-syst-sci-discuss.net/hess-2016-448/). He said it was embargoed! I had to tell him that it was not embargoed, but openly available on the HESS web site, and that I would not wish to comment to a reporter on a study that had yet to undergo peer review. He clearly did not understand the distinction. The paper on the Desmond Storm mentioned above was downloaded over 1000 times, but in the end did not survive peer review. I feel very uncomfortable about this situation.

It seems to me that these rapid assessments submitted to scientific journals are falling between two stools: apparently offering more than what can be said within a few days based on the meteorology of the event and accepted physical principles concerning climate change, but not sufficiently thorough to be a rigorous scientific analysis.

The US NAS extremes report (DOI: 10.17226/21852) recommends "provision of stakeholder information about causal factors within days of an event, followed by periodic updates as more data and analysis results become available". A good model for this is provided by the UK Met Office, who after the January 2014 UK flooding issued a technical report (with CEH) the following month (http://www.metoffice.gov.uk/media/pdf/1/2/Recent_Storms_Briefing_Final_SLR_20140211.pdf). This provided a thorough discussion of the event from a synoptic perspective, with some preliminary discussion of possible links to climate change. It also discussed the flooding, not only the precipitation. It bore the imprimatur of the issuing organizations (and was presumably internally reviewed), so carried considerable weight. The publications in the peer-reviewed scientific literature came only much later, following detailed analysis. That sort of staggered approach seems much more consistent with the NAS recommendations than the paper under discussion here.

---

## Referee Comment (RC2) · Anonymous Referee #2 · 3 Nov 2016

This MS addresses heavy rainfall associated with the may/june 2016 flooding in France and Germany. The paper relies on a method that is available to deploy on short notice to address extreme events in a relatively short turnaround time, in order to have a scientific study supporting media work on the extreme events in question. I have a number of general points and some specific recommendations

1) while the ms is impressive in the breadth and knowledge background it brings to the event, it reads unfinished. There are many questions left hanging in the text, the structure is not great (why have a subsection on a model that isn't even used eg? Wouldn't a sentence do in that case?) and the rushed writing is too visible in the ms to be acceptable in its present form. The figures have strange headings, and not all are

well readable.

2) I am unconvinced about the value of the analysis for Germany given the scale (time and space) of the events. The Germany flooding is due to shortterm highly convective events. Such events are not well resolved in climate models, there are no data to support the hourly extreme resolution involved, raising the question if it really was likely that such an event would have been reasonably well addressed in the models attempted. I wonder if a focus on the France floods wouldnt have made the paper a bit more coherent and readable. Also, summer rainfall may change differently in convection permitting models than parameterized models used here, so there should be quite a bit of caution about that event class a priori.

3) For France, there are interesting differences in results between Seine and Loire between analysis methods and tools, and the reader remains confused about which ones might be statistical flukes and which ones can be explained physically. Also, while the wide array of tools deployed to analyse extreme precipitation in Francis is impressive, there is little scientific explanation of the results. Also, there is little discussion of the synoptic background state and the extent to which the models simulate the key contributors dynamically to those events. This is due to the time constraints. It would be much more satisfying if in addition to a statistical model evaluation an evaluation of the dynamical causes and model ability to resolve it had been possible. I don't understand the Seine results. There is no convincing trend in events with global temperature, yet you find a strong signal. So where might the trend come from? Wouldn't this make you more concerned about reliability? I may well miss the result here because the writing is confusing (see point 1) – but the observational analysis result appears not to be integrated with the model results.

4) It would be very helpful if the model sections were much more integrated among models as well. Models that aren't used can be just mentioned in a sentence rather than have a section. It is also not clear how the models differ in their ability to resolve the events, and a bit of an introduction to that in a more integrated way would be useful,

and what modelling frameworks are particularly suitable and why. The models seem to be selected ad hoc (unsurprisingly based on the timeline but this needs to be resolved in a revised cleaned up ms)

5) It would have been nice to have data for flooding rather than the rainfall per se, as that would have captured also the preconditions of the wet soils as well. On the other hand, I see value in a rainfall analysis as well.

6) I see the questions raised about rapid attribution from the other reviewer and the comment. I am unsure what to recommend here - there is a benefit in being able to address event attribution on a short timescale, although I agree about the limitations by available data and the inability to scrape below the surface of the event is a problem. Some of these issues could be addressed in the review, but with additional data later or additional analysis this also brings the risk of results changing through the review. It does put a burden on the reviewing community as the paper really doesn't read like a polished, finished product. On the other hand, there are several reasons why the results might be reasonably robust: the combination of an observational and model analysis, using multiple models, improves robustness, although the results are not pulled together sufficiently to judge this with confidence (the figures appear to generally show similar results). The author team is fairly cautious about caveats and not over-interpreting results. Being able to have some scientific support for rapid results beats being able to offer only speculation. Also, the relatively large community involved in this specific study helps to make this result more robust, as the approaches used by different researchers and their diverse experience will bring a multitude of perspectives to the event and ensure that key features or problems aren't overlooked. So on balance, I agree with submitting reasonably well researched ms rapidly. However, I think the turnaround time here is too ambitious and an additional few days to week to polish the ms would really have benefitted here. Maybe a format like the BAMs supplement suits the problem better, where a short result could be published quickly followed by detail papers in the scientific literature. In conclusion, I think this study can

be improved to publishable quality, but will require quite much better integration in the revision and clearer addition of caveats, many of which are already implicitly there not well integrated.

Detail comments:

Abstract: l 13 it needs to be added what this estimated return time is based on, so there has to be a bit of the abstract that goes over methods, models and data used.

L 16: why is the precipitation return time more rare for Seine than Loire? Also, this isn't the case in all the models if I interpret the text correctly. What are the return times given here based on – is it the overall summary assessment from all approaches combined?

p. 2 l 18: how do you estimate those return periods? There are a few numbers in the introduction that aren't well supported. On the other hand the breadth of material here is quite impressive and reads interesting.

L 31: low water flows? Or just water flows?

P 3 l 12: in what sense is 3 days the response time of the rivers and how is this determined? (based on basin sizes but how?)

P 3 l 24 Niederbayern is not in the southeastern corner of Bavaria (close though).

Figure 2: what is becaprcp?

p. 5, discussion of findings in literature seem to contradict each other, which isn't discussed well or resolved.

Methods section: why is the my parameter exponential in T – this needs an explanation in the text. Also, given you have 4 parameters to fit, do you have enough data to fit robustly? Lastly, do you include the event in question in your analysis (I suspect not but that should be said, unless I missed it its not there)

L 31 p 6: what is a multiplicative bias correction? It sounds very dangerous here but

later it becomes somewhat more clear what it means and sounds more acceptable but this should be explained here. Also it is not clear in the section above here how the uncertainties are estimated and how and if autocorrelation is accounted for.

p. 7 l 28: I cant see that this is seen in Fig 4bd

Fig. 3 the axis labels are not great! (can be figured out yet. . .). Also wouldn't a longer timeseries in addition be useful? Particularly given that figure 4 shows it only against temperature (and really there is no trend for the Seine!)

The tables are very useful, but should be edited to highlight where results are inconsistent with the data – eg table 1 why are there no results from the observational analysis to compare against? The caption is way too terse to understand it. Which results are significant (highlight bold eg)? Table 2 ceta is really hard to read and many results are just zero compared to observations.

---

## Author Comment (AC1) · 16 Nov 2016

We'd like to begin this response with a brief note of thanks to Professor Shepherd for his comments. Our team is constantly striving to do the best possible science and we take his comments to heart. It is our sincere hope that this point by point response helps provide a better case for how we approach rapid attribution. I summary these show that, to the best of our knowledge, the attribution techniques are state of the art, even comparing to slower analyses,. The use of preliminary data does not affect the outcome much. There is a real demand for such quick scientific attribution information, which we try to provide in an as transparent and rigorous framework as possible. The framework we use at the moment—open review in this journal—may not be optimal,

but we are looking into alternatives whilst rejecting non-transparent options.

*Reviewer 1 has thoroughly identified a number of reasons why this paper is nowhere close to being acceptable as a primary scientific publication, so I won't belabor them but will simply say that I agree:*

*(i) Authors weren't prepared to wait to get access to quality-controlled data;*

We now have the quality-controlled data from Météo France and can compare these with the initial estimates. The maximum 3-day precipitation over the Seine basin in the Météo France Safran reanalysis dataset is 61 mm/3dy and for the Loire 57 mm/3dy. Compared to our initial estimate of 55 mm/3dy and 47 mm/3dy respectively, based on real-time data, we underestimated the rainfall by 10% to 17%. This is well within the uncertainty range given in the original submission and confirms the validity of our previous results, as we show in the reply to reviewer 1. It should be noted that the Risk Ratio is not very sensitive to the severity of the event in general, and here even less as $\xi \approx 0$. So, contrary to the claim, taking real-time data did not materially affect the analysis.

Observations were also used for model validation and trend detection, but as these exclude the event itself they are entirely based on quality-controlled data.

*(ii) Authors weren't prepared to take the time to explore mechanisms for the change in risk, and in particular whether dynamical changes might have been important (which is ironic, since some of the same authors have recently argued that one must do this in event attribution);*

First, the change in risk analysed here is due both to circulation changes and thermo-dynamic changes as the models used in this study simulate this and the observations entail both effects. We have not decomposed the change in risk to thermodynamic and dynamic factors, as this is not a first-order concern to the intended audience (and we never claimed this must be done, but what we did say in the article we think the
commenter referred to, Otto et al. (2016), is the opposite: that **not** reporting the combined effects of thermodynamic and dynamic changes could be problematic). It is an interesting scientific question, which will no doubt be taken up, but does not impact the total change in risk that we wanted to communicate. This is quite standard in the peer-reviewed literature: Schaller et al. (2016) is one of only a few papers that we are aware of where the dynamic component is computed separately. We are just publishing an in depth methodological paper on separating the two effects (Vautard et al., 2016), but this was after the current paper and not yet routine enough to include in a rapid analysis.

*(iii) Although the public concern was about flooding and HESS is a hydrology journal, the authors weren't prepared to take the time to assess the flooding.*

As far as we are aware, there is only one single paper in the peer-reviewed attribution literature that downscales the precipitation to flooding, Schaller et al. (2016), which took more than two years of our time. All other peer-reviewed papers on floods, including all six in the most recent BAMS special supplement, analyse rainfall and do not use hydrological modelling. We judged that an attribution of the rainfall is state of the art at this moment and carefully indicated this in the title. This is of use to the readers of the analysis before the attribution of the floods, which will probably be undertaken but again will probably take a few years.

We did take into account the hydrological situation as much as possible with current tools, by defining the rainfall event in so that a direct link to the impacts (floods) were maintained. In this case by using only the late-spring season so that the relation between extreme precipitation and floods would stay constant, by taking the average over the river basin and by summing over the time scale most relevant to the floods.

We are planning to extend our rapid attributions to flood properties by coupling the output of the climate models to hydrological models. However, this will take a few years of development and testing in slow analyses before the results are robust enough to

include in the rapid ones.

The author of this comment did not mention the strong points of this study, which are rare even in peer-reviewed extreme event attribution articles published on much longer time scales.

1. A careful event definition that does not draw a box around the extreme but sets seasonal, spatial and time boundaries dictated by the impacts.

2. Explicit and quantitative evaluation of the ability of the models used to represent correctly the phenomena being attributed. For France, we reject one of our ensembles of simulations because the PDF does not resemble the observations, even after a multiplicative bias correction. For Germany, we reject most ensembles as these models cannot resolve the relevant spatial scales or have an incompatible PDF. In the end, all models used have a resolution that is high enough to be able to represent the extreme rainfall, in contrast to many peer-reviewed attribution studies that do not include model evaluation or use models that do not resolve the event under study (e.g., Min et al., 2011).

3. Use of multiple high-statistics ensembles of high-resolution models. The results of different climate models that are suitable for the analysis still can differ greatly, especially when simulating summer. A multi-model ensemble gives an indication whether the model uncertainty is larger than the uncertainties due to the weather variability. We are not aware of many studies that use multiple high-statistics ensembles of high-resolution models.

4. An explicit synthesis of the results for France into a consistent attribution statement, and the conclusion that this cannot be done with current information for Germany.

*This particular paper has to be judged on its own merits. However, similar issues arose in the previous HESS submission from this team on the Desmond*

*Storm (http://www.hydrol-earth-syst-sci-discuss.net/hess-2015-534/). The issues would seem to be endemic to the rapid-attribution framework, because they result from the severe time constraints. Thus I would like to take the opportunity that this open review process provides to raise some broader issues about such studies, picking up on the final sentence in Reviewer 1's report.*

*I am concerned that such studies are a disservice to the scientific community. Peer review is the foundation of science, but is done on a voluntary basis. As a former journal editor I am acutely aware of the enormous effort provided gratis by editors and reviewers. To ask them to assess papers that are written so hastily, with so many details left unaddressed, is simply not fair. Time constraints imposed by the media are not a sufficient reason to rush the process of preparing a paper that should meet the standards of rigour expected of an original scientific publication.*

There is no inherent contradiction between rigour and speed. The results of this analysis are as rigorous as most attribution papers, as we indicated by the list above. The main problems with the Desmond paper were insufficient validation and a rushed presentation of our results. We improved upon both these aspects in the current paper, with an explicit model evaluation and an exposition that allows complete reproduction of the results (all the time series are available from climexp.knmi.nl and W@H), again in constrast to much of the published literature. We aim to be as transparent and rigorous as possible.

*Perhaps more importantly, I am also concerned that they are a disservice to the public, and to the public perception of science. There are two aspects to this. The first is that the public wants to know how the extreme event (in this case flooding) was affected by climate change. What they get is a quantitative answer to a different question, based on some proxy for the event (in this case, precipitation over a large region). As the authors are fully aware, the event definition has a very large effect on the quantitative answer, especially when the latter is expressed in terms of return time. So whilst there is a quantitative answer, it is not serving any local resilience need because it is not about*

*the event that captured people's attention. I would call this pseudo-quantification.*

As we mentioned above, we worked hard to make sure that our event definition corresponds as closely as possible to the flood in seasonal, spatial and temporal extent, given the current state of the art in event attribution. We did discuss the flooding on a level similar to the UK analysis the commenter refers to. We hope to move on to a quantitative attribution of runoff or flood levels in the near future. In the only case that we know of that did this, Schaller et al. (2016), it was found that the main factor that changed the attribution significantly was the snow response (less snow in the current climate gave smaller floods). This does not play a role in late spring floods on the Seine and Loire, so we expect there to be agreement between the attribution of the meteorological variable and the hydrological variables. Follow-up studies will show whether this is indeed the case.

*The second aspect is that the Discussion paper looks like a scientific publication, and I suspect the public are unaware that it is not a peer-reviewed publication. I was asked by a reporter (a science writer at Associated Press, so mainstream) to comment on the third paper in this series, on the Louisiana flooding (http://www.hydrol-earth-syst-sci-discuss.net/hess-2016-448/). He said it was embargoed! I had to tell him that it was not embargoed, but openly available on the HESS web site, and that I would not wish to comment to a reporter on a study that had yet to undergo peer review. He clearly did not understand the distinction. The paper on the Desmond Storm mentioned above was downloaded over 1000 times, but in the end did not survive peer review. I feel very uncomfortable about this situation.*

The Associated Press journalist mentioned, in the third paragraph of his story http://bigstory.ap.org/article/3af97f8e1e0c40baab77ffec391dce2a/ noaa-global-warming-increased-odds-louisiana-downpour, emphasised that our analysis was not yet peer reviewed. We have made a point to be very transparent about our process and we have found that journalists do indeed understand the distinction and are careful to report that our results have been submitted for peer
review.

It is important to note that, at the end of his story, the AP reporter writes: 'Most outside experts — including six who contributed to the National Academies of Science report that looked at climate attribution studies — praised the science and results. The national academies panel chairman, retired Admiral David Titley, a Pennsylvania State University meteorology professor, said the Louisiana study followed the guidelines the academies set out and uses observations, models and physics to come to its conclusion. "It's an excellent study," said Columbia University climate scientist Adam Sobel, who was on the academies report team. "These are top established researchers and the GFDL model is one of the best in the business for this purpose. The methods are appropriate and very thoroughly and clearly explained as are the assumptions necessary to draw the conclusions." ' We conclude that there are also many scientists that agree with our approach that did not comment on this HESSD paper.

Concerning the Storm Desmond paper, we have repeated that analysis with all information half a year later, and found indistinguishable results. The numbers that were quoted from it have not changed with all the new information and analysis later. We are documenting this in a follow-up paper in HESS to take away the commenter's 'uncomfortable feeling'. The current paper incorporates the criticism levelled at the Storm Desmond paper: a more complete methods section and explicit model evaluation.

*It seems to me that these rapid assessments submitted to scientific journals are falling between two stools: apparently offering more than what can be said within a few days based on the meteorology of the event and accepted physical principles concerning climate change, but not sufficiently thorough to be a rigorous scientific analysis. The US NAS extremes report (DOI: 10.17226/21852) recommends "provision of stakeholder information about causal factors within days of an event, followed by periodic updates as more data and analysis results become available".*

Indeed, and we try to do as much as possible in the first week or two. Our experience is

that we can compute credible risk ratios in that time frame, which are more informative than making general remarks that may or may not apply in the specific situation. The current paper, with explicit model evaluation, five large high-resolution ensembles and a careful synthesis is already more robust than most of the peer-reviewed literature on extreme event attribution. No doubt in a few years' time our approach will become operational and need not be published. However, at this moment this is not yet the case, as evidenced by the comments on this paper.

We plan to follow up with a more detailed analysis using all information then available, and will investigate as part of that analysis whether the initial computations reported here were accurate. A first look could already be performed at revision time of this paper, carefully noting which information was added later.

*A good model for this is provided by the UK Met Office, who after the January 2014 UK flooding issued a technical report (with CEH) the following month (http://www.metoffice.gov.uk/media/pdf/1/2/Recent_Storms_Briefing_Final_SLR_20140211.pdf). This provided a thorough discussion of the event from a synoptic perspective, with some preliminary discussion of possible links to climate change. It also discussed the flooding, not only the precipitation. It bore the imprimatur of the issuing organizations (and was presumably internally reviewed), so carried considerable weight. The publications in the peer-reviewed scientific literature came only much later, following detailed analysis. That sort of staggered approach seems much more consistent with the NAS recommendations than the paper under discussion here.*

We agree with the commenter that another publication model would be more appropriate than the one we have been using up to now. We are working on this. However, we do not agree that the model discussed above is appropriate. The lack of an open, transparent review process may have been one of the factors that only much later it was found that none of the mechanisms proposed in their section "Weather and climate change drivers" is visible in more than a century of observations (van Oldenborgh et al., 2015; Wild et al., 2015) and the two drivers found in Schaller et al. (2016), thermodynamics and a shift towards more westerly extreme circulation types, were not mentioned.

Our main concern is to publish correct results, and an open and transparent procedure greatly enhances the possibility that results hold up to later scrutiny. Quantitative model evaluation, a multi-model result and a careful synthesis are according to the NAS report necessary to obtain reliable results, and in this paper we perform all of these and thoroughly document the methods and results. The external feedback we have received to date suggests that the science community and the public are served by a timely presentation of these results.

**References**

Min, S.-K., Zhang, X., Zwiers, F. W., and Hegerl, G. C.: Human contribution to more-intense precipitation extremes, Nature, 470, 378–381, doi:10.1038/nature09763, 2011.

Otto, F. E. L., van Oldenborgh, G. J., Eden, J. M., Stott, P. A., Karoly, D. J., and Allen, M. R.: The attribution question, Nature Clim. Change, 6, 813–816, 2016.

Schaller, N., Kay, A. L., Lamb, R., Massey, N. R., van Oldenborgh, G. J., Otto, F. E. L., Sparrow, S. N., Vautard, R., Yiou, P., Bowery, A., Crooks, S. M., Huntingford, C., Ingram, W. J., Jones, R. G., Legg, T., Miller, J., Skeggs, J., Wallom, D., Weisheimer, A., Wilson, S., and Allen, M. R.: The human influence on climate in the winter 2013/2014 floods in southern England., Nature Climate Change, doi:10.1038/nclimate2927, 2016.

van Oldenborgh, G. J., Stephenson, D. B., Sterl, A., Vautard, R., Yiou, P., Drijfhout, S. S., von Storch, H., and van den Dool, H.: Drivers of the 2013/14 winter floods in the UK, Nature Clim. Change, 5, 490–491, doi:10.1038/nclimate2612, 2015.

Vautard, R., Yiou, P., Otto, F. E. L., Stott, P. A., Christidis, N., van Oldenborgh, G. J., and Schaller, N.: Attribution of human-induced dynamical and thermodynamical contributions in extreme weather events, Environ. Res. Lett., to appear, 2016.

Wild, S., Befort, D. J., and Leckebusch, G. C.: Was the Extreme Storm Season in Winter 2013/14 over the North Atlantic and the United Kingdom Triggered by Changes in the West

[Figure]

Pacific Warm Pool?, Bull. Amer. Met. Soc., 96, S29–S33, doi:10.1175/BAMS-D-15-00118.1, in 'Explaining Extremes of 2014 from a Climate Perspective', 2015.

---

## Author Comment (AC2) · 21 Dec 2016

We thank the reviewer for his long and thoughtful comments. It is obvious that rapid attribution, as new scientific activity, provokes a lot of discussion. In hindsight it would have been better to first document our methods in a long normal paper, and build on that for the rapid studies. We attempted to do both in this paper to have a self-contained report of the analysis, but this obviously created a lot of confusion. We plan to follow another format in the future, but still think this is a high-quality attribution study of the specific event in question, the extreme rainfall that led to flooding in France and Germany in May–June 2016. It is innovative in using five high-statistics high-resolution model ensembles, quantitative model evaluation (and rejection), and a thorough synthesis of the attribution results from the observations and different models. This leads to more reliable results than the non-evaluated single model analyses that dominate the literature. We think we have improved the quality of the exposition of the results in a revision enough to merit publication.

Most of the comments on methods noted in the review below would apply to virtually all attribution papers up to now, not only rapid attribution papers, so we do not see why these arguments are used to hold up publication of this study until a unique first-of-a-kind study incorporating hydrology, the role of dynamics and hourly precipitation is ready. It should be noted that the other reviewer is much more positive, and the review of a subsequent paper on the Louisiana floods even notes 'The attribution part of the presented manuscript could be stronger and the group has presented better studies in terms of robustly attributing the role of anthropogenic climate change as this study is primarily based of one model and focused more on a general climatological context than the anthropogenic signal per se. Analyses of the British rainstorm and the French rainfall extremes submitted to the same journal by a similar set of authors better harness the power of multiple methodologies and multi-models.'

We hence feel that it is a valuable contribution to our understanding of the effect of global warming on the class of extreme rainfall events that leads to flooding similar to the observed events. Detailed answers are given below. Addressing the long list of careful comments below has improved the clarity of the manuscript considerably.

1. *First, let me ask the following question: what is the purpose of this "rapid attribution study"? (please note that I am not yet referring to the manuscript) The authors claim that it is motivated by demand on such information: "The extreme nature of this event left many asking whether..." (P1L7), "However demand on such information is often in real time, when, for a couple of days, damages and losses raise the attention to the public and media." (P5L3-4). Consequently, as put by the authors, "A challenge is therefore to provide scientifically sound and*

*reliable information in near real time (about a week) about human influence on extreme events" (P5L4-5). The question here is: who expresses this demand? Is it the general public and the media, as suggested by the authors? If so, what is the actual societal use of delivering such information on such a short time? One may argue that it contributes to the public awareness of the local consequences of anthropogenic climate change by resonating with the short-term memory. I personally find it a weak argument, and I believe that it is not sufficient to drive such a "rapid" study. Conversely, I would perfectly understand performing an attribution study – without the "rapid", as I will detail below – as a way for government, local authorities, or regulators to inform climate change adaptation strategies.*

There are two reasons to do such rapid attribution studies. First, as the reviewer mentions, the public wants to know. As a public servant, should the first author (GJvO) tell the public that we could give them the information they want, of much better quality than the generalities they are given ('everything has changed' or 'we have always had this kind of weather'), but we are not allowed to make this information public? We consider the present study a high-quality analysis, at least equal to many peer-reviewed papers written months or years after the event. Should we just wait for a few months before we submit the exact same paper to avoid accusations of doing science too rapidly? (I know one colleague who did just that after his rapid attribution paper was rejected. The exact same paper was accepted without problems when it was resubmitted a few months later.)

Secondly, there is a lot of evidence from the disaster risk reduction community that important decisions on rebuilding and adaptation after an extreme event are taken within a few months of the event. The information of rapid attribution studies therefore can make these decisions better-informed. Whitty (2015) argues in the health policy context "Since the policy process tends to be very fast, papers must be timely. An 80% right paper before a policy decision is made is worth ten 95% right papers afterwards, provided the methodological limitations imposed by

doing it fast are made clear." We attempt to bring this into practice in climate change attribution with this study.

Indeed, in France, the prime minister has requested a study on the hydrological functioning of the Seine basin for a better adaptation to flood on low flows, with a very short delay, the report was provided in December 2016. The results of the present study were used in this report.

So, both the general public and decision-makers are served by the information in this study, with the second group often obtaining their information from the media as well.

2. *Coming now to the manuscript. It relates this "rapid attribution study", results of which "were completed and released to the public in one week" (abstract, P1L10). The authors have to be congratulated for this performance. However, do scientists really have to be congratulated for performing a study with such a speed? Would the study have been "quick-and-clean", the answer would be definitely yes. But according to the authors themselves, the short time frame imposed severe constraints:*

   - *"However, the data required to analyse the event at the sub-daily scale in real time is not yet available to us (although it is publicly available at DWD)." (P3L31-32)*
   - *"As French precipitation data were not available in real time, the analyses there are based on a relatively sparse subset of stations." (P7L12-13)*
   - *"These values were taken to represent the observed event in the following as the E-OBS data for 29-31 May 2016 was not yet available." (P717-18)*
   - *"We have not yet investigated the reason for this." (P8L16) "We did not managed to process the CORDEX simulations for Germany in the near real time window for the study." (P15L13)*

- *"The methods to answer this question have not yet been developed enough to answer it in the rapid 10-day time window." (P17L12-13)*

We attempt to be honest in describing the limitations of our analysis. Virtually all attribution studies we know of have similar restrictions, sometimes they are just not listed as explicitly. Specifically, answer each of the points above:

- We are not aware of any attribution studies using sub-daily data at all in the published literature. There is an entire EU project (INTENSE) dedicated to collecting and analysing sub-daily precipitation in Europe. When those data are available, the necessary background studies into the quality and homogeneity will become possible. After the strengths and weaknesses have become known we can use these data for attribution studies.

- We now have the quality-controlled data from Météo France and can compare these with the initial estimates. The maximum 3-day precipitation over the Seine basin in the Météo France Safran reanalysis dataset is 61 mm/3dy and for the Loire 57 mm/3dy. Compared to our initial estimate of 55 mm/3dy and 47 mm/3dy respectively, based on real-time data, we underestimated the rainfall by 10% to 17%. This is well within the uncertainty range given in the original submission and confirms the validity of our previous results. It should be noted that the Risk Ratio is not very sensitive to the severity of the event in general, and definitely not to small adjustments like these.

- See above, the CPC estimate proved to be accurate enough for the basin averages.

- Most analyses of observational data do not even consider these questions. We try to flag uncertainties between datasets as a possible limitation of the study.

- We have not decomposed the change in risk to thermodynamic and dynamic factors, as this is not a first-order concern to the intended audience. It is an

interesting scientific question, which will no doubt be taken up, but does not impact the total change in risk that we wanted to communicate. This is quite standard in the peer-reviewed literature: Schaller et al. (2016) is one of only a few papers that we are aware of where the dynamic component is computed separately, albeit only for a single model when it is known that these factors are very model-dependent (see e.g., van Ulden and van Oldenborgh, 2006; van den Hurk et al., 2013). An in depth methodological paper on separating the two effects has just been published (Vautard et al., 2016), but this was after the current paper was written.

*The main issue here is the observational data used. The authors relied on data available in real-time (namely gridded products based on a sparse network of stations), i.e. both sparse and not quality-controlled data. The use of extreme values from such data would at least require checking them against the best available data, which are usually available a month later (for manned rain gauges). The fact that no radar data was used in the study was also surprising as these are usually available in real time, even if their quality may be discussed (but at least they offer a detailed spatial view of rainfields contributing to a robust estimate of catchment-average precipitation).*

As mentioned above, the preliminary numbers we used were very close to the final ones available later and did not materially affect the conclusions. Radar fields are indeed notoriously unreliable until they have been corrected by the manned gauges, which takes a month or more. We tried to minimise the uncertainty by comparing rain gauge estimates with the ECMWF analysis, the agreement between the two gave us confidence to go ahead with the analysis. As noted above, the initial values used in this studies were also found to be good enough for the analysis using the analyses available at revision time.

*A secondary issue here concerns the variable used for this attribution study. In France, it focuses on 3-day precipitation, and not on the actual streamflow values*

*reached during the flooding event. The authors are aware that factors other than this high precipitation intensity came into play:*

- *"[...] many other contributing factors are neglected in this rapid attribution study." (P3L15)*
- *"Firstly the soil types and saturation levels at the time of this extreme rainfall event have not been captured." (P3L15-16)*
- *"This analysis also does not take into account the impacts of the reservoirs [...]" (P3L17-18)*
- *"In addition, land cover and associated runoff characteristics have also not been taken into account" (P3L19)*
- *"A full attribution of the flood themselves, rather than just the rainfall event, would need to take all of these factors into account." (P3L19-20)*

We considered it good practice to mention the limitations of the current study, which are in common with virtually all attribution studies published after flood events. As far as we are aware, there is only one single paper in the peer-reviewed attribution literature that downscales the precipitation to streamflows and flood levels, Schaller et al. (2016). It only considers a single model (neglecting model error) and a single catchment (neglecting all other catchments with losses) and took more than two years of our time. All other peer-reviewed papers on floods, including all six in the 2015 BAMS special supplement, analyse rainfall and do not use hydrological modelling. We judged that an attribution of the rainfall is state of the art at this moment and carefully indicated this in the title. This is of use to the readers of the analysis before the attribution of the floods, which will probably be undertaken but again will take a few years.

*It appears all the more disappointing from the hydrological point of view that such a streamflow attribution study would have been possible (or at least a subset*

[Figure]

*of the experiments) with the help of catchment hydrologists, had the "imposed" time frame not been so short. As a conclusion to this comment, I would ask this question: is the speed at which a scientific study is performed a positive point for evaluating a corresponding manuscript? Can it compensate other negative consequences of the study resulting from the reduced time frame? I will leave the answer to the editor, but from my point of view, this is clearly no.*

Of course such a study is possible. I have no doubt it will come out in a few years' time. Until then, the present study provides very useful information to the public and to decision makers.

It is clear that the hydrosystem reacted to an extreme precipitation event, that is the focus of the present study. Indeed, the 3-day precipitation accumulated on the Loing river basin at Episy represents 1.8 times the volume that discharge on the Loing river at Episy during the 12 days following the events (from May 29 to June 9). For the Seine at Paris, the 3-day precipitation accumulated on the basin represents 2.5 times the observed discharge during the same 12 days that include the flood peak. Thus, even if the hydrological processes that converted and transferred the precipitation event to flood events are of great interest and will certainly be the topic of scientific papers, the focus on the precipitation events itself is important, and is the topic of the present paper.

3. *Coming now to the contents of the manuscript. The authors claim that it took them "an additional week to finalise this article" (abstract, P1L10-11). Again, I believe that the authors should be congratulated provided that their manuscript is clear and sound. However, this is rather not the case:*

- *The manuscript is definitely not well organized: there is no Data nor Discussion sections. Results and discussions and intertwined. Some discussion-relevant elements also appear in the introduction. Results are commented in the conclusions, etc.*

Considering the data section: we decided the manuscript would be much easier to read if the data were introduced at the beginning of each analysis (observations, each model ensemble) separately, rather than first have a long list of observations and model characteristics and next a parallel list of model analyses. This naturally leads to an outline were model-specific results are discussed with this model, and the multi-model results in the conclusions.

- *The central method for deriving risk ratios between now and earlier in the 20th century, and for comparing them with risk ratios derived from factual/counterfactual worlds is not justified nor detailed enough.*

We have expanded the relevant sections greatly. To a good approximation the climate of around 1900 can be considered similar to the counterfactual world without anthropogenic emissions.

- *The text is vague in many locations, on methods, on the use of GCM/RCM data, on the interpretation of results, etc. It therefore makes the study not reproducible at all.*

The methods are documented in section 2. This was rather terse but has been greatly detailed in the revised manuscript. As for reproducibility, all data except Weather@Hone and all methods are publicly available on the public web analysis site climexp.knmi.nl, making reproduction trivial. The Weather@Home data are freely available from climateprediction.net. We greatly value reproducibility in a world were too much data is proprietary and the analyses based on it can never be reproduced by outsiders.

We have greatly expanded and clarified the methods section and the data description in the revision of the manuscript.

- *Several figures are not referred to in the text*
- *There are several inconsistencies between the text and figures/tables*

We have fixed the editorial problems in figure references.

*This manuscript therefore clearly makes the reader feels it has been written in a hurry (like in one week), while it also suggests that the scientific underlying content may be rather valuable (with the restrictions mentioned above). Trying nevertheless to adopt a constructive approach, I took approximately half the time of the authors' writing to identify and list all the points that could/should be improved in the manuscript. This (long) list is given in the Detailed comments section below.*

These are all addressed below.

*In conclusion, I would recommend the editor to reject the manuscript, and to invite the authors to resubmit a manuscript to HESS. I would recommend this new manuscript to be written without the – presumably – artificial time constraints, and to be based on higher-quality observational data that has been made available since their initial submission. I would welcome any further discussion with the authors, the editor, and any other scientific contributor, as I believe the issues raised above are of general significance as they allude to possible drifting ethics in science and science publishing.*

We argue that the revised manuscript provides valuable documentation for a thorough attribution study of the extreme rainfall in France and Germany in May 2016. The reviewer fails to mention the strong points of this study, which are rare even in peer-reviewed extreme event attribution articles published on much longer time scales.

1. A careful event definition that does not draw a box around the extreme but sets seasonal, spatial and time boundaries dictated by the impacts.

2. Explicit and quantitative evaluation of the ability of the models used to represent correctly the phenomena being attributed. For France, we reject one of our ensembles of simulations because the PDF does not resemble the observations, even after a multiplicative bias correction. For Germany, we reject most ensembles as these models cannot resolve the relevant spatial scales or have an incompatible PDF. In the end, all models used have a resolution that is high enough

to be able to represent the extreme rainfall, in contrast to many peer-reviewed attribution studies that do not include model evaluation or use models that do not resolve the event under study (e.g., Min et al., 2011).

3. Use of multiple high-statistics ensembles of high-resolution models. The results of different climate models that are suitable for the analysis still can differ greatly, especially when simulating summer precipitation. A multi-model ensemble gives an indication whether the model uncertainty is larger than the uncertainties due to the weather variability. We are not aware of many studies that use multiple high-statistics ensembles of high-resolution models.

4. An explicit synthesis of the results for France into a consistent attribution statement, and the conclusion that this cannot be done with current information for Germany.

Leaving the present study unpublished gives the signal that the results are unreliable, which they are not. The noted shortcomings in the presentation have been addressed at the revision stage.

**Detailed comments**

Abstract

1. *P1L18 "all four climate models": what are they? Please define clearly the attribution set-up.*

   Changed to 'We evaluated five high-statistics climate model ensembles, four of which simulated the statistics of 3-day basin-averaged precipitation extremes in this May–June well. The results from these four models agree and indicate that the probability of 3-day extreme rainfall in this season has increased by about a factor 2.3 (>1.6) on the Seine and a factor 2.0 (>1.4) on the Loire.'

2. *P1L17 "has increased": between what and what?*

   Added 'over the last century due to anthropogenic emissions of greenhouse gases and aerosols'

3. *P1L19 "The observed trend": over what period?*

   Added 'Over the last century'

4. *Figure 1, a and b: It would be great to have the underlying network of precipitation stations. This should be at least available for E-OBS. And please add a scale to each map.*

   On the contrary, this information is readily available for the CPC product but not for E-OBS, at least not for these months. Fig. 1 of this reply shows that the station density is good enough to trust the basin-wide averages, as also shown by the agreement with later estimates (which has been added to the text, clearly noting that this information was not available at the time of writing but at the time of revision.) We added a sentence to the text 'The showers are also typically smaller than the distance between stations used to construct Fig. 1b, so this map only gives an indication where the heaviest precipitation fell.'

   We added a scale and identifying information to the caption: 'a Precipitation averaged in western Europe (40–60 °N, 15°W–25°E) over 29–31 May 2016 (mm/dy). b Highest 1-day precipitation in Germany and surrounding countries in May 2016 (mm/dy).'

    Introduction

5. *P2L13 "most severe event": in terms of what?*

   Changed to 'flooding event' to make this clearer.

6. *P2L14 "25% of the flood peak of the Seine": How is this estimated? Please provide a reference or a method.*

This estimation is based on the comparison of the volume of the observed discharge of the two basins during the flood peaks.

7. *P2L15 "height above 6.1 m": At what hydrometric station?*

The reference gage for the Seine at Paris is Paris Austerlitz (H5920010). Included in the text.

8. *P2L16-17 "Some measurement problems...": What kind of problems? Please clarify this.*

The river level sensor was not fully effective during the flood period, and a part of data were corrected afterward based on direct observation. This might be not useful to mention this point anymore in the revised version of the manuscript so we deleted this sentence.

9. *P2L19 "of about 20 yr": How is it estimated? Any reference?*

The statistics are provided from http://www.hydro.eaufrance.fr/. The return period was corrected and is now given to be between 5 and 10-year. It was first reference as a 10 to 20 yr return period flood, as can be read for instance in the report of the CCR on the flood (https://www.ccr.fr/documents/23509/29230/Inondations+de+Seine+et+Loire+mai+2016_version+13072016.pdf)

10. *P2L28 "forced to close": on what day?*

RER C was closed from June 2nd until June 10th

11. *P2L28 "without electricity": where?*

The electricity was cut in the flooded area. The extension of the power outage varied in time, and the duration varied in space. Several tens of thousand houses were affected.

12. *P3L2-3 "46% above the previous records": Reference?*

    This estimation is based on the analysis of the observed river flows at Paris Austerlitz. In the observed data set from 1886, the maximum discharge value reached in June was 933 m3/s in 1926.

13. *P3L9 "−40m3.s−1": for a peak flow of...? As such, this figure is not informative.*

    It is correct that the number given may not be informative. We clarify it by stating the reservoirs did not have a strong impact on the Seine flood peak, and provide the reference of the Seine Grands lacs report on the flood. (http://seinegrandslacs.fr/sites/default/files/bilan_crue_juin2016.pdf)

14. *P3L10 "3-day precipitation": Well, this France-averaged estimation is not relevant for all basins. It first heavily depends on the catchment size, but also on flood-generation processes which are catchment-specific. The relevant precipitation time scale for catchments located in the Cévennes area (south-eastern fringe of the Massif Central) is much closer to a few hours, whereas it is several (and usually more than three) days for the Seine@Paris due to the buffering effects of large aquifers. Please better justify your choices here, as the whole study depends on it.*

    The estimate is not valid for France-averaged precipitation, but for precipitation in basins in France. Clarified to 'The meteorological variable that corresponds to the floods on these rivers in France'.

    The 3-day precipitation were an extreme event that get a reaction of the basin. As stated before, the 3-day precipitation accumulated on the Loing river basin at Episy represents 1.8 times the volume that discharge on the Loing river at this gauge during the 12 days following the events (from May 29 to June 9). For the Seine at Paris, the 3-day precipitation accumulated on the basin represents 2.5 times the observed discharge during the same 12 days that include the flood peak. It is correct that the Seine basin is characterised by a strong aquifer that

impact the dynamic of the flood (see for instance Rousset, F., Habets, F., Gomez, E., Le Moigne, P., Morel, S., Noilhan, J., & Ledoux, E. (2004). Hydrometeorological modeling of the Seine basin using the SAFRAN‐ISBA‐MODCOU system. Journal of Geophysical Research: Atmospheres, 109(D14).).

However, the study focusses on the 3-day precipitation that are the main cause of the reaction of the basin, and not on the reaction of the basins that has lasted more that 3 days, (as for instance illustrated by the flood peak on the Seine at Paris that closely follows these 3 days of heavy precipitation).

15. *Figure 2a: What is represented with light blue and orange colours? Please add a legend, and a scale.*

These represent fractional cells, this is now mentioned in the caption.

16. *Figure 2b. Please add a scale and remove the title.*

We have added a scale and removed the title that was inadvertently left on the panel.

17. *Figure 2: The reader should be able to compare the model grid scales and limits (see "which ends at 13eE" in the legend) to the observation network and density. Please add such grid scales in some way to this figure.*

We added the size of the $(0.25°)$ grid to the caption of Fig. 2a. The coloured dots show the box defined in the caption, we judge this is clear enough.

18. *P3L11 "close to the response time": how is it estimated? And on what rivers? With what catchment size? Cf. also above comment.*

It is correct that the sentence was misleading. Indeed, the focus is made on this 3-day precipitation because most of the precipitation fell during this 3-day. It is modified by: 'This major precipitation event was the cause of the flood, the accumulated precipitation being 56% of the total amount during the 16 days of the Seine flood peak.'

19. *P3L31-32 "However, the data required to analyse the event at the sub-daily scale in real time is not yet available to us (although it is publicly available at DWD)": I don't understand. If they are publicly available, why did you not use them?*

First, there is a large difference between the data being available on the DWD web site in their format, and the data being available in a format that the analysis software accepts, especially for station data for which there is no common format (unlike netcdf with CF-conventions for gridded fields). The only reason we can perform these analyses on station data is the availability of GHCN-D and ECA&D data in a standard format on the KNMI Climate Explorer, and the maintainer has not yet found time to expand the site to sub-daily precipitation series.

Secondly, there no studies yet on the reliability and homogeneity of these sub-daily data. We hope to obtain those from the INTENSE EU-project currently underway, so that in the future we can use these sub-daily data.

20. *P5L14 to P6L8: In my point of view, these paragraphs related to future projections are not relevant in the introduction. Results on observed trends and anthropogenic attribution may however be qualitatively checked against findings for 21st century projections in the discussion.*

We removed these paragraphs.

21. *P6L9-14: Please summarise the attribution set-up in a few words. The reader may not understand what kind of models are referred to P6L13.*

Added a sentence: 'We computed the trends in observations since 1950, the trends in SST-forced global climate model simulations since 1960, trends in coupled regional climate models simulations since 1950 and a comparisons with a counterfactual climate without anthropogenic emissions in a large ensemble of SST-forced regional model simulations.'

Methods

22. *Section 2: The whole "Methods" section is quite unclear. Below are (some of the many) points that need clarification, additional references, etc.*

We have rewritten and greatly expanded this section to be clearer. We also moved the description of the synthesis process to this section.

(a) *P6L19 "4-yr smoothed global mean temperature": Please justify the use of the global temperature as an indicator for anthropogenic climate change. For example, why wouldn't you alternatively use the local/regional temperature like in Vautard et al. (2015)? Or maybe the CO2-equivalent greenhouse gas concentrations? This would more in phase with the GCM counterfactual set-up. Less importantly, please also justify the use of a 4-yr smoothing.*

First, the results do not depend noticeably on the choice of covariate, as the measures mentioned by the reviewer are highly correlated. The correlation between annual GMST and the CO2 concentration is 0.93, because in practice the aerosol damping term is also proportional to these. Either will give the same result, given the large natural variability. Using CO2 concentrations would give the impression that these are the only cause. Vautard et al. (2015) also use the smoothed global mean temperature as covariate (the local temperature is employed in the scaling of high precipitation with local temperature on the same day).

We use a 4-yr running mean filter to greatly reduce ENSO variability in the GMST record, which does not impact France very much.

'The smoothing is introduced to remove the fluctuations in the global mean temperature due to ENSO, which are unforced. This measure was already used in van Oldenborgh (2007). (Taking other measures, such as the $CO_2$ concentration or radiative forcing estimates, gives almost the same results as these are highly correlated: for annual means the Pearson correlation coefficient is $r(T', CO_2)=0.93$.).'

(b) *Equation (1): Please define T′. Is it the global mean temperature?*

Yes, added.

**(c)** *P6L25 "values larger than about 0.4 are penalised as unphysical": Please give a reference for documenting and justifying this penalised approach.*

Added: 'An implementation issue is the addition of a penalty term on $\xi$ with a width of $0.2$ so that values larger than about 0.4 are penalised as unphysical. It can be seen from the fits in Figs 4–10 and Table that $|\xi| < 0.1$ for the 3-day averaged basin-wide precipitation. For daily maximum precipitation there are arguments that $\xi \approx 0.12$ Wilson and Toumi (2005); van den Brink and Können (2011). All these values are substantially less than the cut-off. Conversely, time series often have high outliers van den Brink and Können (cf 2008). In the bootstrap procedure, replicating these outliers multiple times gives fits with unphysically high values of the shape parameter. The penalty function does not affect the best fit but keeps these unphysical fits from the sample that is used to estimate the uncertainties.'

**(d)** *P6L26-27 "but take correlations [...] with a moving block technique": Please detail and clarify.*

We have greatly expanded and clarified the procedure.

'When fitting to sets of stations (section 3) or ensembles of model simulations (sections 4–7) we have to take correlations between neighbouring stations or similar ensemble members into account. This is done with a moving block technique analogous to the standard overlapping moving blocks employed when a time series has significant serial autocorrelations (e.g., Efron and Tibshirani, 1998). In that case the block length is set by the time at which the autocorrelation drops to $1/e$. Here, we take bootstrap samples of blocks of stations with correlation $r > 1/e$. In practice this means that after selecting a random year and station, all stations that have a correlation as high as this are also entered into the bootstrap sample, just like a block of years would have been selected in the case of serial autocorrelations. As a

check, we redid the analysis of Vautard et al. (2015) with this technique and verified that we obtained the same result. The same spatial moving block technique was used later in (Eden et al., 2016) and (van der Wiel et al., 2016).'

**(e)** *P6L30 "We evaluate these for the year 2016": I presume this means that you evaluate the cumulative probability density with the temperature of 2016. And so what is this temperature, given that annual temperature for year 2016 is not available? Spring temperature? Please clarify. Same for P7L2-3.*

We used the 3-yr average of 2013–2015 as estimate of the smoothed temperature of 2016. Added.

**(e)** *P7L1: This is actually not a trend detection. This is only a ratio of probability in two specific years, without any formal statistical test, and therefore there is no trend, and no detection. Please rephrase.*

Added 'If this ratio is significantly different from one, i.e., the bootstrapped two-sided 95% confidence interval excludes one, a trend is detected.'

**(g)** *P7L2: Please justify the use of year 1960. Results might have been very different with year 1940 when global temperature first peaked. See also P7L8-9 "the change from 1960 to now...".*

The fit is always over all data available, so the effect of the inclusion of the 1940s depends only on whether the time series go back far enough, not on the choice of reference year (1960 in this case). The choice of 1960 was motivated by keeping the results comparable between the different methods, as some model runs only started in that year.

In this analysis all series start in 1950 or later, but experience in other analyses with longer time series show that the peak makes very little difference in 100+ year long series, i.e., the difference in trends from 1950 and from 1900 is usually much smaller than the uncertainties due to natural variability.

Add 'fit to all available years' to avoid this misunderstanding.

23. *23. Section 2: Reading through this section highlights the fact that a "Data" section is definitely missing before it. Indeed, one cannot grasp the meaning of "neighbouring stations or ensemble members" (P6L27), "(analysis and reanalysis)" (P6L29), "Models" (P6L29), "observed record and reliable models" (P7L1), "two models that also have experiments..." (P7L6-7)*

We have adjusted these to either refer to following paragraphs or made them less specific. Having a separate data section makes the paper significantly less readable.

24. *P7L8-9 "Often we can neglect the effect of natural forcings on these extremes": What does this mean? Please clarify.*

Added: 'We have verified whether this is the case for the extremes in this study.'

Observational analysis

25. *P7L12-13: Again, there should be a map of stations used in the gridded products considered here.*

We feel these are only needed when considering spatial maps of return times (which should never be derived from gridded data unless the station density and grid box size are smaller than the decorrelation scale of the event). In this case, we do the event attribution on three estimates of precipitation averaged over a large region, and have shown before that the consistently of the three estimates gives enough confidence to go ahead with the attribution. We did add that these estimates indeed correspond closely to the official area averages that became at a later time.

26. *P7L14-15 "We checked [...] for the past": I don't understand. What is it about decorrelation scales? Why mentioning ERA-Interim here? Is is used at all?*

We clarified these sentences. 'The decorrelation scales of 3-day precipitation in this season are more than 100 km (derived from the public dense Dutch station

network, we cannot determine how much larger due to the limited size of that country). This is large enough that this is not a problem. We double-checked this by comparing the basin averages with the ERA-interim reanalysis, which is completely independent, and found good agreement.'

27. *P7L15 "Satellite-derived ...": I presume this sentence attempts to justifies the fact that such products are not used here? Please make it explicit and provide some references for their possible low quality of satellite-derived products. And what about radar data? Such products are available in real-time, aren't they? Please provide at least a comment on that.*

The bad performance of satellite rain products is our experience from comparing dozens of time series with co-located satellite series. We added the comparison with E-OBS over the Seine basin, which correlates only at $r=0.7$.

Radar is useful for qualitative analyses of small-scale extremes. It cannot be used for attribution studies because of the limited historical series, O(10 yr), and because of the large errors until it is calibrated against ground-based network, which are often O(30%) and can reach almost 100% when a second shower obscures the first. Added a sentence to this effect.

28. *P7L16: How is the CPC gridded product derived from gauges? Please provide some references, and the grid definition. And what about the temporal homogeneity of the underlying network of stations? I mean, is the list of stations used the same in 1960 than in 2016? This is a critical part of the analysis.*

No, this is not a critical part of the analysis. The CPC series is only used to provide an estimate of the value in 2016, the series is too short to determine a trend from it.

We provided a link to the CPC web site that contains all the requested information. As far as we are aware there is no publication documenting this dataset further. This is also not necessary for this study, because the station density is

far larger than the decorrelation length (see figure above), so any sensible interpolation procedure will give exactly the same results. We have checked that the answer does not depend on the details of the dataset construction by comparing it with the independent ECMWF analysis estimate and found good agreement.

29. *P7L17, Fig. 3: Please use bar charts for plotting precipitation amounts. Plus, what is the smoother line? Climatology? Over what period? Please add a legend. And remove unnecessary text from the figure titles.*

   We have removed the titles, which were meant for internal use only. The smooth line is indeed the climatology, added this to the caption. We will consider bar charts for future publications, but at this moment we do not know how to plot a bar graph with climatology.

30. *30. P7L21 "The trend analysis": There is no trend analysis in Eq. 1. Please clarify.*

   Clarifies to 'The GEV fit using Eq. 1, which includes a trend analysis by fitting the trend parameter $\alpha$, '

31. *P7L21-22 "larger than can be determined with the longest series": First, what does this mean? Second, what is the longest series?*

   The text reads 'larger than can be determined with the longest series, E-OBS'. To make this even clearer we have expanded the text. to "the longest series, which is E-OBS".

   To determine a return time $\tau$ one needs a series of at least length $\sim \tau/2$ years. All observational series we have access to are shorter than this. Added ' (Note that a fit to an extreme value distribution can only determine a return time smaller than about $2N$ yr from a series of length $N$ yr with any accuracy.) ' to the text for readers unfamiliar with extreme value analysis.

32. *P7L23 "The best fit for the trend is positive": Please rephrase.*

    'The best fit for the trend parameter $\alpha$ is positive'

33. *P7L23 "However, the uncertainties are large and easily encompass zero": zero what? Are these uncertainties related to the red vertical bars that appear in (but are not commented nor even mentioned in the legend of) Fig. 4a?.*

    No, they are unrelated. Changed 'zero' to '$\alpha=0$., i.e., no trend'

    Added 'The vertical bars indicate the 95% confidence interval on the location parameter $\mu$ at the two reference years.' to the caption.

34. *P7L25-26 "We can improve the estimate": What do you mean exactly by "improving"? Please clarify.*

    ''We can reduce the uncertainties in the estimate of the return time'

35. *P7L25-27 and Fig. 4b: This is really confusing. Precipitation values mentioned above (P7L16-17) are a sum over 3 days, the plot title of Fig. 4c) suggests a daily average, the plot presumably shows the daily average, and the legend says "3-day rainfall". I would strongly suggest that all plots are made with 3-day sums, not to confuse the reader with the alternative possibility of studying one-day extremes.*

    We have changed the earlier paragraph to also quote 3-day averaged values, in order to minimise the possibility of introducing errors in changing everything else, and in order not to use the confusing units 'mm/3dy'. We also went through the text and changed '3-day' to '3-day mean' wherever necessary. It should be noted that all quantities of interest in this study, return times and changes in return times, do not depend on whether we take the mean or sum.

36. *P7L25-27 and Fig. 4b: Another highly confusing point: the text mentions that the fit is eventually made with a Gumbel distribution. However, the legend of Fig.*

*4 says that what is plotted is a GEV fit. Please clarify. What adds again to the
confusion is the unsuited use of the term "Gumbel plot" for GEV fits...*

For the model evaluation we need the GEV fit, which is shown in Fig. 4.

As far as we are aware the term 'Gumbel plot' is the standard term for a plot on
which the X-axis has been transformed to $log(log(\tau))$ so that a Gumbel curve is a
straight line. It does not imply that the data is being fitted to a Gumbel distribution.
In fact, it is the standard way to present a GEV fit. Changing standard terminology
into a non-standard one would confuse most readers even more, the term "return
time plot' does not imply which transformation has been used for the X-axis.

We added the figures of the Gumbel fits to support this paragraph.

37. *P7L32-33 "The number [...] recently": This is confusing. What is the number of
stations in 1951 for example? Please clarify and rephrase.*

'The number of active stations increases sharply to 219 in 1951, stays in the
range 220–240 from 1951 to 2000 and decreases to about 190 recently.'

38. *P8L2-5: This is clearly not enough supported by details on the procedures or
relevant references. Please provide some more details/references.*

The spatial moving block bootstrap technique is now described in great detail,
including references, in the methods section (see above).

'We therefore analyse all April–June maxima together as in Vautard et al. (2015),
starting in 1951 to minimise the inhomogeneity due to the start of the modern
network and taking spatial dependencies into account as described in section 2.'

39. *P8L6: What is the "present climate"? Please clarify.*

' the present climate (red lines, Eq. 1 with the smoothed global mean surface
temperature set to the current value) '

40. *P8L7-8 "The probability [...] this area": This statement is not supported by any figure, or am I wrong? Please clarify.*

   'The probability of observing this at any station in the region is of course much higher, about 23 yr (8 to 1800 yr, fit not shown), as these showers of $\mathcal{O}(10km)$ are small compared to this area of $\mathcal{O}(400km)$.'

41. *P8L9-10 "... so the spatial heterogeneity". This is unclear, please rephrase.*

   'As the study area includes significant orography (e.g., the Black Forest, Schwäbische Alb, Fichtelgebirge) the results could be influenced by spatial heterogeneity of the stations, i.e., that they do not have the identical distributions assumed in the fit. One way to test this is to normalise all series to the same mean of annual 3-day extremes in April–June and repeating the analysis. This gives a RR of 0.5 (0.4 to 0.8), leading us to conclude that spatial heterogeneity does not affect this result.'

42. *P8L15 "The re-analysis": What reanalysis? ERA-Interim? I thought you were discussing about E-OBS? Please clarify.*

   We started a new paragraph to indicate that this is indeed a different discussion.

   Sections 4 to 8

43. *General comment for these sections: model runs should be presented (period, grid, forcings) beforehand in the data section. And Sections 3 to 8 reorganized in a Results section.*

   We tried this, and found it made the paper considerably harder to read than in its present form, with model description, results and discussion for that model alone grouped in individual section. This is the information that the reader needs to have together to judge the results. We combine the results in the synthesis section.

[Figure]

44. *P9L4: Please specify what are the forcings.*

   'based on the CMIP5 historical forcings up to 2005 and RCP4.5 afterwards'

45. *P9L7: How do you deal with the ensemble of realizations in the fitting procedure? I presume you put all realizations together in order to reduce the sampling uncertainty. Please clarify.*

   Indeed. ' The simulations were made as an ensemble of 15 realisations for the historical forcings, and another ensemble of 15 realisations for the historicalNat forcings. The 15 members were all entered into the fit simultaneously. For this variable the series are sufficiently independent ($r<1/e$) in spite of the common SST forcing.'

46. *P9L7 "for the same forcings": Please rephrase.*

   See above.

47. *P9L7-8 "The data... use.": Wouldn't it be better suited in the acknowledgement section with a contact or website?*

   We consider it important for reproducibility to mention it here, together with the analysis. Too many papers are based on data that is not publicly available.

48. *P9L1-8: What is the spatial resolution of HadGEM3-A? Again, please make a map of the model grid. This is critical to justify the comparison with observed (well, gridded) data.*

   As mentioned in the first sentence of this section: N216, about 60km. We added a map similar to Fig. 1a with the HadGEM3-A grid.

49. *P9L9-10 "The model ... (table 2)": Once again, this is really confusing. Is this statement valid for both forcings? Is it relative to the location parameter $\mu$? If yes, I can't see how you may obtain the figures mentioned in the text from Table 2.*

The historicalNat runs can by definition not be compared to the observations, so we thought it was clear that this referred to the historical runs. Added 'Comparing the annual maximum of 3-day mean basin-averaged precipitation of the historical runs to the E-OBS observations (excluding 2016), '

Table 2 gives $\mu = 9.2$ for the best fit of the HadGEM3-A data, $7.8$ for E-OBS. The difference is rounded to 15% when all decimal places are taken into account, we do not want to suggest more accuracy than is warranted on the basis of the large uncertainties.

50. *P10L3: Table 2 only shows a difference in the probability of occurrence of 55 mm/3dy between 1960 and 2016, not an "increase in extreme precipitation". Please rephrase.*

    This is not correct. Table 2 shows the parametrisation of the tail of 3-day precipitation as well as the probability of occurrence of 18.4 mm/dy. This parametrisation $\mu, \sigma, \xi, \alpha$ of Eq. 1 describes all extremes.

51. *P10L5 "at p < 0.025": What is the statistical test used? Again P10L13.*

    This is derived from the bootstrap that has been described in the methods section. Added this.

52. *P10L8-9: Again there is no trend here. Only a difference between estimates for two different years.*

    This is not correct. The fit of Eq. 1 describes a GEV with a trend $\alpha$, which is sampled at the two different years in the figure. The trend parameter $\alpha$ is compatible with zero. Added the intermediate step in the reasoning for clarity.

53. *P10L10-12 "2.0 [...] (0.6 to 7.2)": Please detail how these figures are obtained.*

    Clarified to 'Comparing the historical and historicalNat return times in the current climate'. The rest of the procedure should be obvious: the risk ratio is obtained by dividing the two return times and propagating the errors.

54. *P10L13-15: This is unclear. Please rephrase.*

    'However, the trend in the historical runs is significantly different from zero. Conversely, the RR is near one in the historicalNat runs, hence natural forcings do not give rise to a trend. Finally the RR from that analysis also agrees well with the one obtained from the difference between historical and historicalNat runs. These three points are evidence that the trend is mostly due to anthropogenic emissions.'

55. *Figure 6: This figure is not referred to in the text. In the legend the model acronym is not consistent with the text and tables.*

    Added the reference.

    Inserted the missing dash in the titles to make them consistent.

56. *P11L2 "CMIP5 protocol": Please detail this protocol. I presume this is the historical runs until 2005. What about afterwards?*

    Added 'historical and RCP8.5'. According to Kirtman et al. (2013), the difference between the different RCPs is negligible up to about 2030.

57. *P11L6-11: This overall negative assessment is really interesting and useful to the community. Same comment for P13L19-21.*

    Thank you. We consider it essential that models are evaluated before using them in an attribution study and hope the current paper will establish this as standard (if it is published).

58. *P12L3: Please make it clear what is the difference between the first experiment (Climatology) and the other two.*

    The text already mentions that 'Climatology' refers to 1986–2014 and 'Historical' and 'HistoricalNat' to 2014, 2015 and 2016. We are unable to make this any clearer.

59. *P12L10-11 "The 2016 data": I don't understand. Please clarify and detail all data types (variables, etc.) used and their specific purposes for the study.*

Modelled zonal 200 hPa wind anomaly data (w.r.t. 1986–2014) for April–June 2016 are used to diagnose the potential dynamic contribution that current SSTs might have had on the European floods, such as a lagged effect of the strong 2015/16 El Niño event. It enables us to estimate the change in likelihood of the event occurring as a result of to the anomalous circulation in comparison to the climatological mean circulation during the same time period. This result can then be contrasted with the change in risk due to thermodynamically driven, warming related modifications of the background atmosphere.

60. *P12L12 "The availability ... attribution." Could you give some examples?*

Added citations to some relevant analyses: '(e.g., Otto et al., 2012; Schaller et al., 2014; Uhe et al., 2016)'

61. *P12L13-14 "how it depends ... Eq. 1)": This highlights the lack of comments on that point (noted in an above comment) when introducing Eq. 1.*

We added a paragraph detailing the two underlying assumptions to the description of Eq. 1 and how we check them:

'After fitting Eq. 1 to data we verify that the underlying assumptions are not invalid. Specifically, the return time plots show whether the distribution can be described by a GEV by overlaying the data points and fit for the present and a past climate. Deviations, such as caused by a double populations, are clearly visible on this plot. The second assumption, that the PDF scales with the smoothed global mean temperature, is checked in the high-statistics Weather@Home model. The high number of data points means that the extremes in that model can be studied without these assumptions.'

62. *P12L17: Please stick to RR once you defined it. Again P13L4, P13L5.*

Shortened.

63. *Figure 7: The legend mentions "dots", but plots show also lines. What do they represent? I presume it is an envelop of the individual members, but why not pooling them (as for HadGEM, if I am right)? Please clarify.*

As mentioned in the legend, the central dot is the return time, the bar the 95% confidence interval. Attempted to clarify this further. 'Red dots are return times for current conditions ('historical'/ACT) with the horizontal lines denoting 95% confidence intervals'.

64. *Figure 7: Please display the observational value for 2016 in the plots.*

We explain in the text why we do not do this. Instead, vertical lines at the lower and upper boundaries of the 95% CI of the return time from the observational analysis are now shown.

65. *P13L6-7 "Note that ... other analyses.": I don't understand. Please clarify.*

'Note that the similarity of the curves in Fig. 7 and in the other figures justifies the assumptions made in the other analyses. The first is that the distributions are described well by a GEV (also verified in each plot by the quality of the fit to the data points). The second one, which can only be checked here, is that this GEV scales with global warming. There no indications that the difference between the red and blue curves in Fig. 7 is different from the other model analyses (Figs. 6, 8, 9) beyond the uncertainties indicated by the 95% error bars.'

66. *P13L8: What is GloSea5?*

The UK Met Office seasonal forecast system, see Haustein et al. (2016). Changed to 'seasonal forecast SSTs'

67. *P13L8-18: This should belong to a discussion part.*

It has been moved to the discussion.

68. *P13L9 "We found ... over Europe.": Could you explain?*

Added: 'In other words, the zonal wind anomalies in 200 hPa do not indicate that there was an increased tendency for an event like this to occur in central Europe as 2016 upper level winds did not deviate significantly from the climatological mean (1986–2014) in the model.'

69. *P13L10-11 "However ... event": Again, this is not clearly enough explained.*

Clarified to: 'However, this does not mean that there is no case-specific contribution as summer circulation anomalies are fairly weak in general anyway and extreme weather usually driven by other factors. In fact, the climatology experiment (black dots in Fig. 7) does suggests a strong role for case-specific dynamic contributions in case of the Seine event, though less so for the Loire event.'

70. *P13L12 "and Seine run-off.": There are presumably missing words in the sentence. If this aims at suggesting that Seine runoff is higher than normal in post-Niño years, please provide a reference.*

added 'higher'. There was a publication 15 years ago showing this, but I am afraid I cannot remember the authors nor find it back in my literature database nor in on-line databases. There is a correlation of $r=0.39$ between Apr–Jun Seine run-off at Paris and Niño3.4 two months earlier, we added that instead.

71. *P13L25 "CORDEX": First use of the term. Please define.*

I am afraid this is an acronym which is much better-known than the underlying expansion. The CORDEX home page does not have it. Expanding it does not make the text any more readable, but rather less readable.

72. *P13L26 "internal variability": Please specify that this is EC-Earth (and not RACMO2) internal variability.*

[Figure]

This is not the case. Especially in the summer half year, the RCM also generates internal variability on top of the internal variability of the driving GCM, as the typical scales of weather in that season are smaller than the domain used.

73. *P13L31 to P14L1: Please make sure that bias values are consistent with results shown in Table 2 and Table 3.*

The difference between the RACMO and E-OBS estimates of $\mu$ is compatible with zero within the $2\sigma$ error bounds when considering the unrounded numbers, whereas for the Loire zero just falls outside the $2\sigma$ interval. We agree this is to some extent arbitrary, but some boundary has to be drawn. In the end, it makes very little difference to the analysis.

'The GEV fit parameters for the Seine are compatible with the fit to the observations (Table 2,3) within the $2\sigma$ uncertainties, so we accept the model for this analysis and do not apply a bias correction. In the Loire region the model results are just significantly different, about 5% lower than the observed ones, so we do apply this small bias correction. The distribution in Germany is also similar to the corresponding gridded observations (E-OBS, see Table 4), albeit somewhat wetter, so we accept this model there as well with a 15% bias correction.'

74. *P14L1-2: Please refer to Table 4.*

Thank you, see above.

75. *P14L3-5: Please refer to Fig. 8. This Figure is not referred to in the manuscript.*

Thank you, added.

76. *P14L4-5: Again, what is the statistical test? What does "very significantly" mean?*

This is again counting the number of bootstrap members for which the trend is zero or negative, which is now described in the methods section. With 'very significant' we try to indicate that there are none in a 1000-member bootstrap, so

the $p$-value is very low, but the ensemble is too small to accurately state how low. Added $p<0.01$ to stay on the safe side.

77.  *P14L6-7: Please refer to Fig. 9. This Figure is not referred to in the manuscript.*

     Thank you, added.

78.  *P14L12: Please justify the use of runs forced by RCP8.5. I know this may have very little influence, but I'd like it to be commented.*

     Again added a reference to the relevant IPCC WG1 AR5 chapter.

79.  *P15L4 "biases": I presume on precipitation?*

     'These all have different biases in the annual maximum of 3-day mean precipitation averaged over the river basins. '

80.  *P15L4 "simple scaling to a common mean": This is the second time in the paper that this procedure is referred to, but it is still unclear what is this common mean. Do you simply divide by the observed mean? In that case, at what spatial resolution? Please clarify.*

     Added 'for which the mean of all simulations is taken'.

81.  *P15L6-7 "The uncertainties take that into account": Well, this is far from being sufficient as an explanation, and far from being reproducible. Please detail.*

     The procedure has been detailed in the methods section (even more in the revised version). It works exactly the same for dependent station data as it works for dependent ensemble members, as indicated in these sentences. It is automatically implemented by the routines on the Climate Explorer that were used for these analyses, but a post-doc here implemented them in R without any problems and obtains the same results, starting from this description. This indicates that reproducibility is not as bad as the reviewer indicates, either using the publicly

available routine on the Climate Explorer website, or implementing it independently.

Added 'as detailed in section 2'.

82. *P15L8 "The basin averages over the Seine and Loire": I presume you mean "the distribution of basin-average April-June 3-day maximum precipitation"... Please try and be more accurate.*

Thank you, changed to 'The distribution of the annual maximum of April-June 3-day mean basin-average precipitation over the Seine and Loire'.

83. *P15L8-12: Please refer to Fig. 10. This Figure is not referred to in the manuscript.*

Added.

84. *P15L10-11 "This is ... other models": Is it a formal statement or a more qualitative one?*

This sentence has been cut, as it should be discussed in the next section.

Conclusions

85. *P16L2 "Floods on the Seine are rare this time of the year" and P16L2 "only two late spring/summer floods have been recorded before in over 500 years": Well, I disagree factually. Out of the 30 remarkable flood events identified for the EU Floods Directive (EU, 2007) in the "Seine-Normandie district", eight occurred during April to June (Lang and Cœur, 2014, p. 386): April-May 1983, 16-17 June 1997, April-May 1998, 7-13 May 2000, March-April 2001, 1 June 2003, 7-8 June 2007, 14 June 2009. Among these only, already two show a very similar pattern of soil saturation followed by intense rain, on areas close or very close to those hit by the 2016 rainfall event: the 10-15 April 1983 flood particularly hit the Essonne subcatchment (Lang and Cœur, 2014, p. 404-405), the April-May 1998 flood hit the Yonne and Loing (mentioned P2L7 for 2016) subcatchments (Lang*

*and Cœur, 2014, p. 415-416). But there is also (for example) the 16-23 March 1978 flood that hit the small tributaries south of Paris, including the Yvette river mentioned P2L7 (Lang and Cœur, 2014, p. 403-404).*

The sentence was misleading since the focus was on the Seine river only, not on the Seine basin, and not upstream the Seine river but in Paris. It is now corrected to 'Major floods on the Seine river are rare this time of the year. Although the overall return time of the flood crest at Paris was about 20 years, only two late spring/summer floods have been recorded there in over 500 years before 2016.' The analysis is based according to the records from the regional agency for environment and energy (www.driee.ile-de-france.developpement-durable.gouv.fr).

86. *P16L7 "The observational records are too short to establish a trend over the last 65 years.": I don't understand. Is 65 years too short a period to derive a robust trend? As for the length of observational records, the precipitation series available from Météo-France over the Seine basin allow for a computation of basin-scale daily average as least as reliable of this from E-OBS, and for a much longer period. Please rephrase.*

Unfortunately, the Météo France data are not publicly available and hence could not be used in this study. We hope that this limitation will be removed in future collaborations including Mété France, e.g., in EUPHEME, or by a transition to a more open data policy, as has happened in Germany and other European countries.

Added this to the text: 'The observational records available to us…'.

87. *P16L10-12: This should belong to the Methods section. Plus, this is rather unclear as such.*

We have added paragraphs to the Methods section describing the synthesis in much more detail, and refer to that here..

88. *P16L14-15 "We just ... result": Please rephrase and detail.*

We now refer to the new detailed description in the methods section.

89. *P16L14 to P17L2: Some of this should also belong to the Methods section.*

It is now described here in detail.

90. émphTable 1: Why does it appear as Table 1 as it is only commented after the two other ones? Plus, I presume you meant "natural-2016" on row 8 column 4.

The other tables were meant as appendices, as they are referred from all model sections. This one should be close to this discussion. We leave the ordering to the editor.

We have standardised the headers to preind., thank you for noting this.

91. *P17L3-13: This belongs to the Discussion section.*

This is the discussion section, now made explicit in the title.

92. *P18L5-7: I am not sure that comparing the trends in extreme precipitation values in Germany with that of the Cévennes range (with high orographic effects) and Jakarta (in a tropical setting with monsoon influence) is necessarily relevant...*

The Black Forest goes to 1493 m, the Cévennes to 1702 m, so the difference is not that large. We left out Jakarta and added a recent article on the extreme precipitation in Boulder (Eden et al., 2016), which found very little trend there.

93. *P18L11 "the two analyses": What are they?*

Observational and RACMO, added.

**Technical corrections**

All these have been addressed.
*1. P1L14: "return time" → "return period", and throughout the whole manuscript. This is the most commonly used terminology in hydrology.*
*2. P1L16: "once roughly" → "roughly once"*
*3. P1L17 "Seine a factor": probably a missing word*
*4. P2L2: "rainfalls" → "rainfall"*
*5. P2L22: "less" → "lower"*
*6. P2L22: I believe you mean "March 2016'*
*' 7. P2L23: the official name is "EU Sequana 2016"*
*8. P2L11: "seizes" → "sizes"*
*9. P7L3 and P8L12: "ration" → "ratio'*
*' 10. P7L16: "55,mm" → "55 mm"*
*11. P12L13: "assumptions to" → "assumptions on"*
*12. P16L2: "flood crest" → "flood peak"*
*13. P16L14 "his" → "its"*
*14. P18L10 "his" → "this"*
*15. P23L2: "precipitations" → "precipitation" C17*

**References**

Eden, J. M., Wolter, K., Otto, F. E. L., and van Oldenborgh, G. J.: Multi-method attribution analysis of extreme precipitation in Boulder, Colorado, Environ. Res. Lett., 11, 124 009, doi: 10.1088/1748-9326/11/12/124009, 2016.

Efron, B. and Tibshirani, R. J.: An introduction to the bootstrap, Chapman and Hall, New York, 1998.

Haustein, K., Otto, F. E. L., Uhe, P., Schaller, N., R., A. M., Hermanson, L., Christidis, N., McLean, P., and Cullen, H.: Real-time extreme weather event attribution with forecast seasonal SSTs, Environ. Res. Lett., 11, doi:10.1088/1748–9326/11/6/064 006, 2016.

Kirtman, B., Power, S. B., et al.: Near-term Climate Change: Projections and Predictability, in: Climate Change 2013: The Physical Science Basis, edited by Stocker, T. F. et al., chap. 11, pp. 953–1028, Cambridge University Press, Cambridge, U.K. and New York, U.S.A., 2013.

Min, S.-K., Zhang, X., Zwiers, F. W., and Hegerl, G. C.: Human contribution to more-intense precipitation extremes, Nature, 470, 378–381, doi:10.1038/nature09763, 2011.

Otto, F. E. L., Massey, N., van Oldenborgh, G. J., Jones, R. G., and Allen, M. R.: Reconciling two approaches to attribution of the 2010 Russian heat wave, Geophys. Res. Lett., 39, L04 702, doi:10.1029/2011GL050422, 2012.

Schaller, N., Otto, F. E. L., van Oldenborgh, G. J., Massey, N. R., Sparrow, S., and Allen, M. R.: The heavy precipitation event of May–June 2013 in the upper Danube and Elbe basins, Bull. Amer. Met. Soc., 95, S69–S72, 2014.

Schaller, N., Kay, A. L., Lamb, R., Massey, N. R., van Oldenborgh, G. J., Otto, F. E. L., Sparrow, S. N., Vautard, R., Yiou, P., Bowery, A., Crooks, S. M., Huntingford, C., Ingram, W. J., Jones, R. G., Legg, T., Miller, J., Skeggs, J., Wallom, D., Weisheimer, A., Wilson, S., and Allen, M. R.: The human influence on climate in the winter 2013/2014 floods in southern England., Nature Climate Change, doi:10.1038/nclimate2927, 2016.

Uhe, P., Otto, F. E. L., Haustein, K., van Oldenborgh, G. J., King, A., Wallom, D., Allen, M. R., and Cullen, H.: Comparison of Methods: Attributing the 2014 record European temperatures to human influences, Geophys. Res. Lett, doi:10.1002/2016GL069568, 2016GL069568, 2016.

van den Brink, H. W. and Können, G. P.: The statistical distribution of meteorological outliers, Geophys. Res. Lett, 35, L23 702, doi:10.1029/2008GL035967, 2008.

van den Brink, H. W. and Können, G. P.: Estimating 10000-year return values from short time series, Int. J. Climatol., 31, 115–126, doi:10.1002/joc.2047, 2011.

van den Hurk, B. J. J. M., van Oldenborgh, G. J., Lenderink, G., Hazeleger, W., Haarsma, R. J., and de Vries, H.: Drivers of mean climate change around the Netherlands derived from CMIP5, Clim. Dyn., doi:10.1007/s00382-013-1707-y, 2013.

van der Wiel, K., Kapnick, S. B., van Oldenborgh, G. J., Whan, K., Philip, S., Vecchi, G. A., Singh, R. K., Arrighi, J., and Cullen, H.: Rapid attribution of the August 2016 flood-inducing extreme precipitation in south Louisiana to climate change, Hydrol. Earth Syst. Sci. Disc., pp. 1–40, doi:10.5194/hess-2016-448, 2016.

van Oldenborgh, G. J.: How unusual was autumn 2006 in Europe?, Clim. Past, 3, 659–668, doi:10.5194/cp-3-659-2007, http://www.clim-past.net/3/659/2007/, 2007.

van Ulden, A. P. and van Oldenborgh, G. J.: Large-scale atmospheric circulation biases and changes in global climate model simulations and their importance for climate change in Central Europe, Atmos. Chem. Phys., 6, 863–881, doi:10.5194/acp-6-863-2006, 2006.

Vautard, R., van Oldenborgh, G. J., Thao, S., Dubuisson, B., Lenderink, G., Ribes, A., Soubey-

roux, J. M., Yiou, P., and Planton, S.: Extreme fall 2014 precipitations in the Cévennes mountain range, Bull. Amer. Meteor. Soc., 96, S56–S60, doi:10.1175/BAMS-D-15-00088.1, 2015.

Vautard, R., Yiou, P., Otto, F. E. L., Stott, P. A., Christidis, N., van Oldenborgh, G. J., and Schaller, N.: Attribution of human-induced dynamical and thermodynamical contributions in extreme weather events, Environ. Res. Lett., 11, 114 009, doi:10.1088/1748-9326/11/11/114009, 2016.

Whitty, C. J. M.: What makes an academic paper useful for health policy?, BMC Medicine, 13, 301, doi:10.1186/s12916-015-0544-8, 2015.

Wilson, P. S. and Toumi, R.: A fundamental probability distribution for heavy rainfall, Geophys. Res. Lett., 32, L14 812, doi:10.1029/2005GL022465, 2005.
* * *
[Figure]

prcp 29−31May CPC daily precipitation

**Fig. 1.** Station density of the CPC dataset in May 2016